# Atom of Thoughts for Markov LLM Test-Time Scaling

**Fengwei Teng**[1,2],   **Quan Shi**[3],   **Zhaoyang Yu**[2],   **Jiayi Zhang**[1,2],
**Yuyu Luo**[1],   **Chenglin Wu**[2†],   **Zhijiang Guo**[1†]
[1]HKUST(GZ),   [2]DeepWisdom,   [3]Renmin University of China

## Abstract

Large Language Models (LLMs) have achieved significant performance gains through test-time scaling methods. However, existing approaches often incur redundant computations due to the accumulation of historical dependency information during inference. To address this challenge, we leverage the memoryless property of Markov processes to minimize reliance on historical context and propose a Markovian reasoning process. This foundational Markov chain structure enables seamless integration with various test-time scaling methods, thereby improving their scaling efficiency. By further scaling up the Markovian reasoning chain through integration with techniques such as tree search and reflective refinement, we uncover an emergent atomic reasoning structure, where reasoning trajectories are decomposed into a series of self-contained, low-complexity atomic units. We name this design Atom of Thoughts (AoT). Extensive experiments demonstrate that AoT consistently outperforms existing baselines as computational budgets increase. Importantly, AoT integrates seamlessly with existing reasoning frameworks and different LLMs (both reasoning and non-reasoning), facilitating scalable, high-performance inference.We submit our code alongside this paper and will make it publicly available to facilitate reproducibility and future research.

## 1   Introduction

Large Language Models (LLMs) exhibit remarkable scaling behavior: as model parameters and training data increase, their performance improves predictably across a wide range of tasks [21]. Recently, test-time scaling methods have emerged to push the performance boundary further by increasing computational resources during inference. These range from basic Chain-of-Thought (CoT) prompting that extends reasoning chains [33], to more structured approaches like Tree-of-Thought (ToT) [40] and Graph-of-Thought (GoT) [4] that organize multiple LLM invocations for exploring solution spaces, and recent reasoning models such as OpenAI O1 [26] and DeepSeek R1 [9] that enhance LLMs' long-chain reasoning ability through post-training [29, 24, 18].

However, current framework-based test-time scaling methods typically rely heavily on retaining extensive historical information. Even the simplest CoT must preserve the entire reasoning trajectory to generate each subsequent step [33, 53]. Tree-based methods maintain ancestor and sibling relations for branching decisions [40, 56, 11], while Graph-based methods introduce even more complex dependencies through arbitrary node connections [4, 52]. Figure 1b analyzes these representative structures and abstract the complexity of historical information and reasoning completion token involved at each LLM invocation.

To decouple the current problem's reasoning from processing historical information and thus minimize their mutual interference during test-time computation, we aim to generalize Markov chain–style structures to general-purpose reasoning. By exploiting the **memoryless property** of Markov processes, we design the Markovian reasoning process, where each state encapsulates a self-contained

---

[†]Corresponding Authors. Contact: steamedbun2002@outlook.com

39th Conference on Neural Information Processing Systems (NeurIPS 2025).

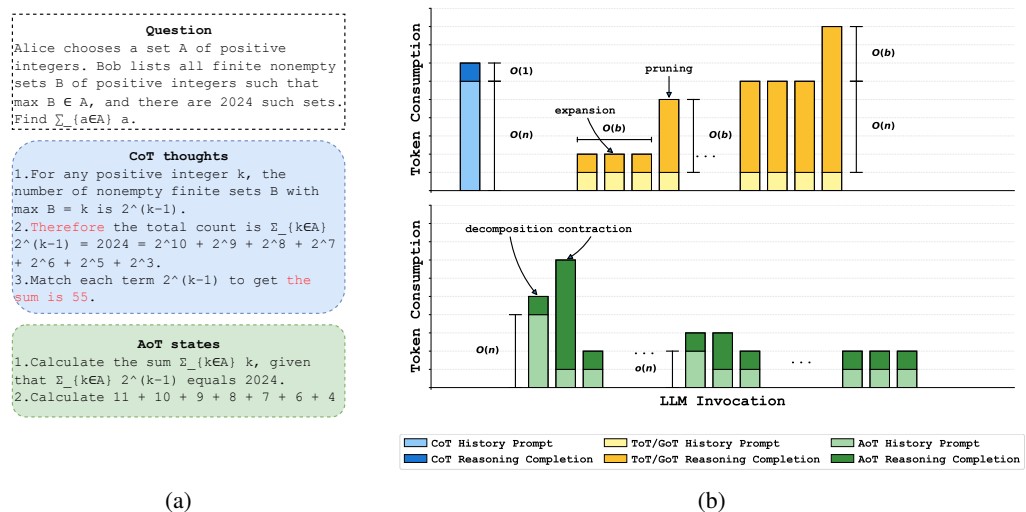

(a)                 (b)

Figure 1: **Token Allocation Comparison in Reasoning Frameworks.** *Figure (a)* demonstrates the differences between thoughts and states, where the red-highlighted text in thoughts reflects dependencies on historical information, whereas states maintain answer-equivalence with the initial problem while progressively reducing execution complexity. *Figure (b)* illustrates differences in the number of prompt tokens and completion tokens for CoT, ToT, GoT, and the state-based AoT. For simplicity, we assume each thought consists of the same number of tokens, with an average of $O(n)$ thoughts required to express a solution. While ToT maintains $b$ branches, resulting in a fixed number of $b$ invocations per expansion stage, GoT's settings can be flexibly adjusted depending on the scenario and are thus denoted as $O(b)$.

problem, thereby significantly reducing historical dependencies. The reasoning process is expressed as a sequence of states with progressively reduced test-time complexity, rather than an accumulation of historical thoughts like CoT, as illustrated in Figure 1a. To ensure steady progress, we introduce a two-phase state transition mechanism: the decomposition stage converts the current state into a Directed Acyclic Graph (DAG)-based reasoning path, and the contraction stage uses its structure to reduce dependencies and generate the next state.

This fundamental structure with memoryless property distinguish our approach from many CoT-based methods. Thus, our method can be seamlessly integrated with existing test-time scaling methods, enhancing their scaling efficiency. While exploring integrations with tree search and reflective refinement to further scale up the Markovian reasoning chain, we identify an emergent trend towards an atomic reasoning structure (Figure 4), where reasoning trajectories are represented as a series of self-contained, low-complexity atomic problems. To emphasize this characteristic, we name our approach Atom of Thoughts (AoT).

Our contributions are summarized as follows:

- **Markovian Reasoning Process.** We introduce a general-purpose Markovian reasoning process that achieves high-quality and cost-effective reasoning across various scenarios, including code generation, mathematical reasoning, and multi-step reasoning tasks.

- **Scalable Reasoning Structure.** The basic structure design of Markov chain in AoT facilitates seamless integration with various test-time scaling methods, significantly enhancing computational efficiency and allowing the combination of different methods' advantages. This scalability ensures more effective utilization of increased computational budgets without the overhead of maintaining extensive historical contexts.

- **Atomic Reasoning.** Further leveraging AoT's seamless integration capability to enhance itself, by integrating with tree search and reflective refinement to scale up the exploration of the Markovian reasoning chain, we uncover an emergent atomic reasoning structure. In this structure, complex reasoning trajectories are decomposed into a sequence of atomic, self-

contained units with low complexity. This atomicization brings about improved reasoning performance and robustness.

## 2 Related Work

### 2.1 Reasoning Framework

Drawing inspiration from cognitive behaviors in human reasoning [3]—such as step-by-step decomposition [33, 57, 31, 13], reflective refinement [23, 55, 54], and aggregation ensemble [32, 20, 42]—various prompting strategies have been developed to enhance the reasoning capabilities of LLMs. These reasoning frameworks typically employ structured representations, including chains, graphs, and trees [40, 4, 51], to model the reasoning space efficiently and systematically. Chain-based methods, for instance, decompose complex problems into linear sequences of subproblems [33, 57, 31], primarily optimizing for stepwise dependency. In contrast, tree- and graph-based formalisms support hierarchical exploration of multiple reasoning paths, allowing for more dynamic adaptation during the problem-solving process [40, 4]. These structured approaches have demonstrably improved LLM performance in diverse applications like code generation, question answering, and complex data processing [17, 16, 48, 50], by enabling LLMs to tackle intricate problems with enhanced coherence and interpretability.

While these structured methods significantly expand LLMs' reasoning capabilities, they also inherently accumulate historical dependencies. This accumulation can lead to increased computational costs and potential interference during the inference process. Recent efforts have attempted to mitigate this reliance on historical information by exploring Markovian reasoning processes and atomic reasoning steps, aiming for more memoryless transitions [36, 34, 58, 35]. However, these approaches often suffer from task-specific design limitations, hindering generalizability and efficient parallelism [12, 38, 46]. In contrast, AoT introduces a DAG-based approach that decouples partial subproblems into atomic nodes. This decoupling enables independent state transitions without the substantial overhead associated with maintaining historical context. By iteratively decomposing problems into these atomic nodes and then contracting them, our method reduces overall complexity and inherently supports efficient parallel execution, thereby addressing the limitations of traditional chain, tree, and graph-based structures.

### 2.2 Test-Time Scaling

Test-time scaling has emerged as a powerful mechanism to enhance LLM reasoning by extending computational effort during inference. Framework-based approaches augment LLM capabilities through structured reasoning extensions, leveraging cognitive operations and external tool integration to facilitate deeper exploration of solution spaces [49, 28, 7]. These methods introduce reflective reasoning cycles, recursive problem-solving, and dynamic path selection, significantly improving performance on complex reasoning tasks. Despite these advances, existing techniques commonly preserve full historical state information throughout the reasoning process. This can lead to redundant computational overhead and potential conflicts across successive reasoning steps.

Recent work has explored alternative strategies, such as supervised fine-tuning on CoT trajectories, demonstrating improved LLM capacity to maintain coherent, long-term reasoning [44, 6, 41]. Reinforcement learning have further pushed these boundaries by enabling models to autonomously extend reasoning chains, potentially unlocking emergent cognitive patterns [22, 47, 9, 45]. However, similar to framework-based methods, these techniques often rely on maintaining expansive historical contexts, which can limit their efficiency and scalability as reasoning paths become extended.

In contrast to these history-dependent methods, our approach adopts a Markovian perspective, modeling the reasoning process as state transitions assisted by a temporary DAG structure. This memoryless design eliminates the need for redundant history tracking, focusing computational resources solely on current state transformations. Furthermore, our proposed two-phase transition mechanism, comprising decomposition and contraction stages, facilitates atomic problem-solving. This enhances computational efficiency while maintaining structural clarity. This structured yet flexible approach not only reduces dependency overhead but also aligns naturally with the principles of test-time scaling, offering seamless integration with existing reasoning frameworks to achieve scalable, high-performance inference.

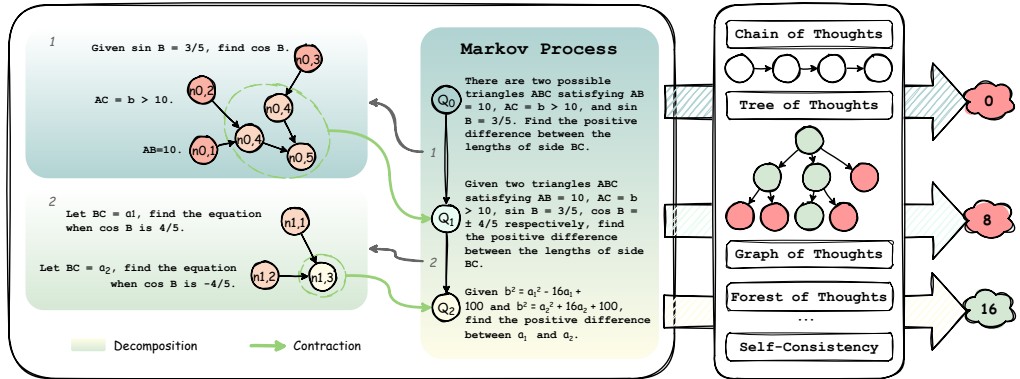

Figure 2: **Overview of AoT.** The Markov reasoning framework iteratively derives states $Q_{i+1}$ from predecessors $Q_i$ using DAG decomposition and contraction. The left part shows this iterative process, while the right part highlights the integration with existing methods. Any intermediate state $Q_i$ can act as an entry point $Q_0$ for other methods, ensuring flexible composition while preserving answer equivalence to the original question. This allows AoT to operate independently or as a preprocessing module to optimize the performance or efficiency of existing approaches.

# 3 Atomic Reasoning via Markov Process

In this section, we first formally derive a Markovian reasoning process grounded in a clear probabilistic formulation. We then discuss how this Markovian reasoning structure can be integrated seamlessly with other reasoning methods to further scale up inference time. Finally, we demonstrate how atomic reasoning structures naturally emerge through such scaling-up procedures. The overview of this Markovian reasoning process is illustrated in Figure 2.

## 3.1 Markovian Reasoning Process

**Reasoning Chain.** CoT reasoning introduces a sequence of intermediate steps $T_i$ to solve a problem. This process can be formalized as a probabilistic sampling procedure:

$$A \sim p(A|\mathcal{T}, Q_0) \prod_{i=0}^{N} p(T_i|\mathcal{T}_{<i}, Q_0) \tag{1}$$

where $A$ is the final answer, and $\mathcal{T} = \{T_0, T_1, \ldots, T_N\}$ is the sequence of thoughts, each conditioned on the previous steps $\mathcal{T}_{<i}$ and the initial question $Q_0$.

An alternative formulation—Least-to-Most [57] prompting—reframes the node of chain as a sub-question $Q_i$, yielding:

$$A \sim p(A|\mathcal{Q}) \prod_{i=0}^{N} p(Q_i|\mathcal{Q}_{<i}) \tag{2}$$

Under the above formulation, the reasoning process is characterized by the accumulation of intermediate thoughts or subquestions in the sequence, leading to a continual increase in historical information. However, ideally, if the reasoning chain satisfies the property of a memoryless Markov process—where each state $S_{i+1}$ depends only on $S_i$—we obtain:

$$A \sim p(A|S_N) \prod_{i=0}^{N} p(S_{i+1}|S_i) \tag{3}$$

where $S_i$ represents a state in the Markovian reasoning process. In the following paragraph, we will explicitly clarify the semantic content of the Markov state $S_i$, resulting in a more specific and practical representation.

**Markov State.** In practice, real-world problems rarely satisfy the strict Markov assumption directly. To establish a meaningful Markovian formulation, we reuse the subquestion symbol $Q_i$ to represent the Markov states $S_i$, initialized by the original question $Q_0$. Since the final answer $A$ must be derivable from the final state $Q_{-1}$, it follows naturally that $Q_{-1}$ is answer-equivalent to $Q_0$. Thus, an essential invariant emerges: each intermediate subquestion $Q_i$ must preserve answer-equivalence with the original question. To ensure meaningful Markov state transitions, we further impose that the sequence of subquestions $\{Q_0, Q_1, \ldots, Q_N\}$ monotonically reduces in complexity, guaranteeing genuine reasoning progress at each transition.

**Two-phase Transition** However, state transitions aiming at test-time reduction remain challenging for LLMs, especially without task-specific training. This difficulty arises primarily from the complex historical dependencies within reasoning trajectories. To address this issue, we propose a two-phase transition mechanism that first explicitly decomposes the current state $Q_i$ to capture the internal dependencies before contracting them into the next state.

In the decomposition phase, we introduce a DAG scaffold $\mathcal{G}_i$ to explicitly represent the dependency structure among reasoning steps within each intermediate question $Q_i$. This temporary structure is later discarded to eliminate historical dependencies, enabling the Markovian transition. Formally, the DAG is defined as:

$$\mathcal{G}_i = (\mathcal{N}, E), \quad E \subseteq \{(N_j, N_k) \mid j < k\} \tag{4}$$

where nodes $N_k$ represent individual thoughts or subquestions, and edges $(N_j, N_k)$ indicate that node $N_j$ provides necessary information for node $N_k$.

In the subsequent contraction phase, we transform the temporary DAG structure $\mathcal{G}_i$ into the next Markov state $Q_{i+1}$. Specifically, nodes without incoming edges in $\mathcal{G}_i$ are independent and can be safely discarded, whereas the remaining dependent nodes are reformulated into an answer-equivalent independent question $Q_{i+1}$. Formally, the overall Markovian transition process can be expressed as:

$$A \sim p(A|Q_N) \prod_{i=0}^{N} p(Q_{i+1}|\mathcal{G}_i) \, p(\mathcal{G}_i|Q_i). \tag{5}$$

A detailed step-by-step example demonstrating the complete decomposition-contraction process is provided in Appendix B.2.

## 3.2 Emerged Atomic Reasoning

The Markovian reasoning process provides a fundamental, low-level structural prior for inference. In this subsection, we discuss the design of a termination mechanism to counteract the potential fragility introduced by strict memorylessness, thereby constructing a stable reasoning framework. Moreover, we describe how this Markovian reasoning structure can be combined with additional methods—particularly through structured exploration via tree search and reflective verification—to further scale up test-time reasoning. This combined approach reveals the emergence of a stable, indivisible reasoning structure, termed atomic reasoning.

**Termination Strategy.** Unlike CoT-based approaches, which can recover from early errors by leveraging accumulated context, our Markov chain lacks such a fallback due to its memoryless nature. This amplifies the risk of propagating low-quality transitions—if an intermediate question $Q_{i+1}$ diverges semantically from the original task, subsequent reasoning becomes meaningless.

To address this, we introduce a quality-aware termination strategy. After each transition $Q_i \rightarrow Q_{i+1}$, an LLM-as-a-judge selects the best answer to the original question $Q_0$ from the triplet $\{\text{solve}(Q_i), \text{solve}(\mathcal{G}_i), \text{solve}(Q_{i+1})\}$. Crucially, this mechanism implicitly enforces answer equivalence: if $Q_{i+1}$ fails to preserve answer equivalence with $Q_0$, then $\text{solve}(Q_{i+1})$ will not provide a valid answer for $Q_0$ and thus cannot be selected by the judge. This selection-based filtering naturally ensures that only semantically stable transformations maintaining answer equivalence are retained. If $Q_{i+1}$ is not selected, the process terminates and returns the best candidate among the three. Detailed quality metrics demonstrating the effectiveness of this mechanism are provided in Appendix B.1.

**Modular Integration.** Since each Markov state is constrained to be an equivalently transformed representation of the original question, the reasoning process forms a semantically aligned and

Table 1: Performance Comparison.

| Model | Benchmark | CoT | CoT-SC | SR | AR | AFlow | ToT | GoT | FoT | AoT |
|---|---|---|---|---|---|---|---|---|---|---|
| **Non-Reasoning LLMs** | | | | | | | | | | |
| GPT-4o-mini | MATH | 78.3 | 81.8 | 78.7 | 65.4 | 83.0 | 82.0 | 82.3 | 82.6 | **83.6** |
| | GSM8K | 90.9 | 92.0 | 91.7 | 87.2 | 93.5 | 91.8 | 92.1 | 94.2 | **95.0** |
| | MBPP | 72.4 | 73.2 | 72.8 | 70.1 | 74.0 | 73.5 | 73.7 | 74.8 | **75.2** |
| | LongBench | 57.6 | 58.6 | 58.2 | 52.9 | 61.0 | 59.0 | 59.2 | 60.8 | **68.5** |
| DeepSeek-V3 | MATH | 94.4 | 95.2 | 94.8 | 90.1 | 96.1 | 95.0 | 95.3 | 95.6 | **96.5** |
| | GSM8K | 96.2 | 97.0 | 96.8 | 92.5 | 97.8 | 96.5 | 96.8 | 97.5 | **98.2** |
| | MBPP | 75.7 | 76.5 | 76.0 | 73.2 | 77.3 | 76.8 | 77.0 | 78.2 | **79.6** |
| | LongBench | 58.8 | 60.1 | 59.5 | 55.3 | 63.5 | 61.2 | 61.5 | 63.3 | **71.0** |
| **Reasoning LLMs** | | | | | | | | | | |
| O3-mini | AIME | 79.6 | 81.0 | 80.2 | 76.0 | 82.5 | 81.2 | 81.5 | 81.8 | **83.0** |
| | LiveCodeBench | 23.6 | 25.0 | 24.2 | 20.0 | 26.5 | 25.2 | 25.5 | 27.8 | **32.2** |
| | LongBench | 56.3 | 57.5 | 56.8 | 52.0 | 58.0 | 56.5 | 56.8 | 57.9 | **65.3** |
| DeepSeek-R1 | AIME | 78.3 | 79.7 | 78.9 | 74.7 | 81.2 | 79.9 | 80.2 | 80.5 | **81.7** |
| | LiveCodeBench | 24.5 | 25.9 | 25.1 | 20.9 | 27.4 | 26.1 | 26.4 | 28.1 | **30.9** |
| | LongBench | 55.1 | 56.2 | 55.4 | 52.3 | 58.7 | 57.0 | 57.5 | 58.2 | **67.9** |

fully self-contained sequence of problem representations. This property enables modular reasoning without compromising the integrity of the overall task. In practice, each state within the chain can be independently routed to specialized solvers, subjected to verification procedures, or further embedded into structured reasoning frameworks—such as tree-based or graph-based inference. The introduction of the Markov reasoning process thus does not merely offer an alternative to previous reasoning chain methods, but rather defines a structural foundation upon which diverse test-time reasoning strategies can be constructed.

**Atomic Structure.** Although the termination strategy ensures robustness, it also restricts the emergence of deeper reasoning chains. To explore the full potential of the Markov process, we sample and extend trajectories, combining tree search and reflection mechanisms. These structured explorations reveal a statistically supported phenomenon: deeper reasoning states tend to converge into irreducible forms, maintaining a stable and relatively low reasoning token count, from which the original problem's answer can be directly inferred with high execution stability. We refer to these stable forms as atomic structures: indivisible and self-contained representations that require no further decomposition. Importantly, atomicity is not imposed a priori, but emerges naturally as a property discovered throughout the reasoning process. This convergence toward atomic units represents a logical endpoint where problems become sufficiently simple that further decomposition is neither necessary nor beneficial. Notably, this convergence point is jointly determined by both the intrinsic complexity of the problem and the reasoning capabilities of the underlying model—different problems may converge at different depths, and the same problem may exhibit different atomic granularities when solved by models with varying capacities.

# 4 Experiments

Our experiments aim at two primary objectives. First, we conduct main experiments across a variety of datasets spanning mathematics, code generation, and multi-hop question answering to demonstrate the cost-efficiency advantages of AoT as a general-purpose reasoning framework. Second, leveraging the flexibility provided by the basic Markov chain structure in our approach, we design integration experiments at various granularities. These experiments explore the utilization of AoT as a plug-in component to enhance cost-efficiency in other reasoning frameworks and investigate scaling effects in integration with classical methods like tree search and verification-based reflection, analyzing emergent reasoning phenomena.

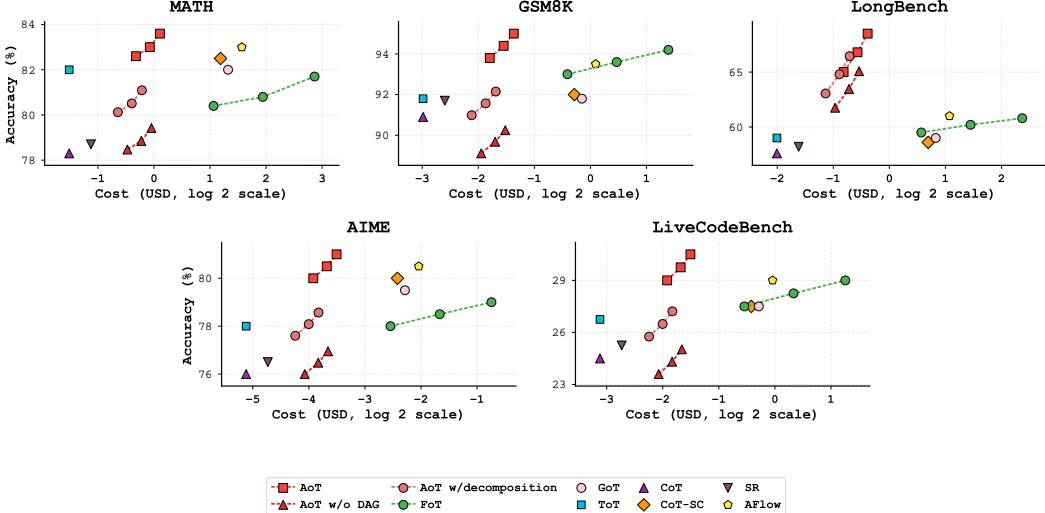

Figure 3: A comparison of performance and cost of various methods and ablation methods on the dataset, with GPT-4o-mini as the backbone. Each node in the curves represents an AoT (or ablation variants) iteration result, where increasing token consumption indicates deeper iterations. Due to relatively poor AR performance leading to scattered data points, AR data points are excluded.

## 4.1 Experimental Setup

**Benchmarks and Metrics.** We evaluate AoT across representative benchmarks covering mathematical reasoning (MATH [14], GSM8K [8], AIME[1]), code generation (MBPP [1], LiveCodeBench [19]), and multi-hop question answering tasks (HotpotQA [39], MuSiQue [30], and 2WikiMultiHopQA [15] preprocessed by LongBench [2]), see Appendix F.3 for details. Following previous work [49, 5], we report pass rates for mathematical and coding benchmarks, and F1 scores for multi-hop QA tasks.

**Settings.** All prompt templates used in Markov reasoning process for experiments are fully described in Appendix A.1. Key hyperparameters, including model temperature and Markov chain length, are detailed and discussed in Appendix A.2. We set the default temperature to 1.0 and the maximum Markov chain length to 3 for the main experiments to balance performance and efficiency while enabling scaling curves. Due to AoT 's design and termination mechanism, longer chain lengths increase the performance ceiling without linearly increasing costs.

**Backbones and Baselines.** AoT is designed to be compatible with various LLM backbones. To demonstrate its effectiveness, we employed two categories of LLMs. The first category comprises non-reasoning LLMs, specifically GPT-4o-mini [25] and DeepSeek-V3 [10]. The second category includes reasoning-capable LLMs such as O3-mini [27] and DeepSeek R1 [9]. Specifically, we use non-reasoning models to evaluate performance on MATH, GSM8K, and MBPP, and reasoning-capable models to evaluate performance on more challenging tasks such as AIME and LiveCodeBench. Additionally, since multi-hop QA is not a primary focus for reasoning-capable models, both categories of models are evaluated on LongBench for comprehensive comparison.

For comparison, we evaluated AoT against a diverse set of baseline methods, broadly categorized by their interaction pattern with the LLM: single-call or multi-call invocations. Single-call approaches include well-known techniques like Chain-of-Thought (CoT) [33] and Chain-of-Draft (CoD) [37]. Multi-call methods represent more complex workflows, such as CoT with Self-Consistency (CoT-SC) [32], Self-Refine (SR) [23], Analogical Prompting (AP) [43], Forest-of-Thought (FoT) [5], and the agentic framework AFlow [49]. Further details are provided in Appendix A.3.

---

[1]https://huggingface.co/datasets/Maxwell-Jia/AIME_2024

## 4.2 Main Results

Table 1 presents the main experimental results. Across both Non-Reasoning and Reasoning LLMs, AoT consistently demonstrates strong performance. For Non-Reasoning LLMs such as GPT-4o-mini and DeepSeek-V3, AoT achieves the highest scores on benchmarks like MATH, GSM8K, MBPP, and LongBench, often surpassing all other compared methods. For instance, with GPT-4o-mini, AoT scores 83.6 on MATH, 95.0 on GSM8K, 75.2 on MBPP, and 68.5 on LongBench, which are the top performances. Similarly, DeepSeek-V3 with AoT leads with scores on all benchmarks.

In the Reasoning LLMs section, featuring O3-mini and DeepSeek-R1, AoT continues to exhibit competitive and often leading performance. For O3-mini, AoT achieves the highest scores on AIME (83.0), LiveCodeBench (32.2), and LongBench (65.3). With DeepSeek-R1, AoT again leads on all tasks. Overall, AoT consistently achieves state-of-the-art or highly competitive results across a diverse set of models and benchmarks, demonstrating its effectiveness.

Figure 3 further demonstrates that performance improves progressively with additional reasoning iterations. This highlights the effectiveness of our proposed termination strategy: by mitigating error propagation from memoryless Markovian transitions, it preserves the desirable test-time scaling property—performance does not degrade as more computational resources are allocated.

## 4.3 Ablation Study

We conduct ablation studies to examine the impact of core components in our framework. Specifically, we evaluate two variants: (1) Without Decomposition, where the model directly contracts reasoning trajectories from the initial question without constructing a DAG; and (2) Without DAG-guided Contraction, where decomposition still occurs, but the contraction step does not rely on any structural guidance. In this setting, only the first naturally independent subproblem is separated out. Figure 3 shows that both ablations significantly degrade performance, with the second variant causing a more severe drop. This suggests that partial or superficial structural cues can be more harmful than providing none at all. These results underscore the importance of explicitly modeling fine-grained dependencies in reasoning trajectories, showing that faithful structural representations meaningfully enhance reasoning effectiveness and precision. Comprehensive quality metrics for the DAG generation process, including answer equivalence maintenance rates (>99% across all datasets) and complexity reduction rates (74-82%), are provided in Appendix B.1.

## 4.4 Scaling Up Analysis

In this section, we further explore the scalability of AoT by integrating it with existing reasoning frameworks, leveraging its flexible, modular design. We begin our analysis by using individual Markov states as integration points—a lightweight and straightforward approach where intermediate states processed by AoT serve as optimized entry points for other reasoning methods. Our experiments reveal substantial efficiency improvements at test-time, which encourages us to examine larger, more structured integration granularities to fully capitalize on the structural strengths of our framework. Notably, as we progressively extend the Markov chain during scaling analysis, we observe a consistent reduction in the number of tokens required for reasoning in the final states. Through detailed analysis, we identify emerging atomic characteristics in the reasoning trajectories, motivating us to design further scaling-up experiments based on this property.

**State Integration.** The Markov states $Q_i$ generated by AoT represent simplified, yet answer-equivalent reformulations of the original questions, making them ideal entry points for external methods. Indeed, AoT itself demonstrates such modular integration potential, employing basic CoT-style prompting to solve each intermediate state. To experimentally validate the effectiveness of these intermediate states, we investigate whether initiating reasoning using optimized intermediate states $Q_1$ can enhance both accuracy and computational efficiency in external frameworks. The results, illustrated in Figure 4, confirm that starting reasoning from these optimized intermediate states notably improves performance while simultaneously reducing computational costs, as demonstrated in the integration with frameworks such as FoT.

**Tree Searching.** Beyond single-state integration, the full Markov sequence $\mathcal{Q}$ generated by AoT can provide a structured scaffold for more complex reasoning frameworks, effectively replacing

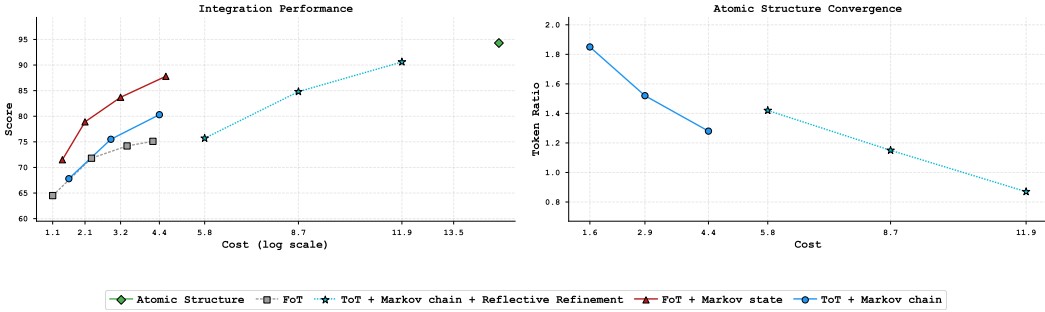

Figure 4: The process involves gradually enhancing integration for scaling up at test time. ToT uses three branches, while FoT employs two, four, and eight trees, respectively.

traditional CoT-based structures. In conventional CoT-based ToT, the inherent randomness of LLM-based sampling can lead to inconsistencies in reasoning chain lengths, causing nodes at the same depth to represent varying stages of reasoning progress. This inconsistency complicates node comparison and diminishes pruning effectiveness. In contrast, the Markov chains constructed by AoT ensure answer equivalence between each intermediate node and the original question, thereby guaranteeing fundamental comparability across nodes at the same depth. This structural consistency significantly enhances the gains from scaling through parallel sampling at test-time.

**Reflective Refinement.** Termination strategy in AoT provides a safeguard for the quality of single-pass Markov reasoning. When a transition yields a low-quality intermediate state, early termination allows the system to avoid wasting computation on unpromising paths. However, this conservative mechanism may also limit further exploration. To address this, we augment our method with verification-based reflection, where transitions $Q_i \rightarrow Q_{i+1}$ are evaluated by an LLM-as-a-judge to assess whether the newly generated state exhibits a significant degradation in test-time performance. If such degradation is detected, the system triggers a reflective refinement step, encouraging deeper and more meaningful reasoning rather than trivial reformulations. This reflective verification substantially improves comparability between nodes at the same depth, increases the effective exploration space, and further amplifies the benefits of structural scaling. When combining all three integration strategies (ToT + Markov chain + Reflective Refinement), we observe significant performance gains: for instance, on MATH, this full integration achieves 84.9% accuracy compared to ToT's 82.0%, and on AIME, it reaches 81.2% versus ToT's 78.0%, demonstrating the compounding benefits of our modular design.

**Atomic Struture.** Due to the inherent scalability of the AoT architecture, deeper Markov chains—enabled by both tree search and verification-based reflection—exhibit stronger test-time performance and require fewer reasoning tokens in the final state. Statistical analysis reveals that the token count of final reasoning steps gradually approaches that of a minimal DAG representation comprising all independent subproblems generated during transitions. This suggests a natural convergence toward atomic states—questions that are semantically represent indivisible reasoning units. We refer to this phenomenon as atomic reasoning, where the entire reasoning trajectory is composed of such minimal, non-decomposable elements. To further validate this insight, we conduct an additional experiment where we isolate and re-execute these highly atomic reasoning paths independently. While this incurs significantly higher computational cost, the results exhibit stable scaling trends, highlighting the structural advantages of AoT with high budget.

## 5 Conclusions and Future Work

We present AoT, a general-purpose reasoning framework that leverages Markovian transitions to minimize historical dependencies during inference. By alternating between decomposition and contraction, AoT incrementally reduces complex queries into atomic subproblems, enabling scalable and modular reasoning across maths, code, and multi-hop QA tasks. Empirically, we show that AoT not only scales gracefully with compute but also integrates flexibly into existing reasoning paradigms as a plug-in module. Limitations and broader impacts of AoT are provided in Appendix **??** and **??**.

While AoT offers a promising path toward atomic reasoning, its current implementation operates solely at inference time. A natural extension is to align this structure with training-time objectives—teaching models to internalize Markovian and atomic reasoning patterns directly. This could involve supervised fine-tuning with synthetic traces, reinforcement learning over decomposition trajectories, or pretraining on datasets that promote context-isolated reasoning.

More broadly, this work lays the foundation for reasoning systems that emphasize minimal context, compositionality, and structural modularity. We hope AoT serves as a stepping stone toward more efficient, interpretable, and robust reasoning with large language models.

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

## Appendix Overview

This appendix is organized into three main parts: Section A provides comprehensive implementation details including prompts, hyperparameters, and baseline configurations; Section B presents detailed empirical analyses validating our framework's effectiveness; and Sections **??–??** discuss limitations and broader impacts of this work.

## A    Implementation Details

This section provides comprehensive implementation details necessary for reproducing our experiments, including prompt templates, hyperparameter settings, and baseline method configurations.

### A.1    Prompt Templates

We present the core prompt structures used in AoT for different task domains. Our framework employs four key prompt types: (1) `direct` for solving problems, (2) `decompose` for extracting DAG structures, (3) `contract` for generating simplified questions, and (4) `judge` for LLM-as-a-judge evaluation. Below we detail domain-specific implementations for mathematical reasoning, code generation, and multi-hop question answering.

**Design Rationale.**    The Multi-hop QA prompts use JSON for structured responses, while Math and Code tasks use HTML-like tags (e.g., `<answer></answer>`). This design choice reflects task-specific requirements: JSON naturally accommodates Multi-hop QA's need for structured outputs including reasoning chains and supporting evidence, while HTML tags provide clear answer demarcation for Math and Code tasks. Function parameters also vary by domain—Multi-hop QA requires context passages, Code generation needs test cases and dependency information, while Math tasks only require the question. These variations align with the inherent characteristics of each problem type rather than representing arbitrary design choices.

### A.1.1    Mathematical Reasoning

```python
def direct(question: str):
    instruction = """
        You are a precise math question solver. Solve the given math
    question step by step using a standard algebraic approach:

        QUESTION: {question}

        You can freely reason in your response, but please enclose the
     final answer within <answer></answer> tags (pure number without
    units and explanations)
    """
    prompt = instruction.format(question=question)
    return prompt

def decompose():
    instruction = """
        Decompose the previous reasoning trajectory into a series of
    sub-questions or thoughts.

        Instructions:
        1. Each sub-question or thought should list its other sub-
    questions or thoughts' indexes it depends (0-based, can be an
    empty list)
```

```
        2. Dependencies are defined as information needed in sub-
    question or thought that:
            - Does NOT come directly from the original question
            - MUST come from previous sub-questions or thoughts
    """
    return instruction

def contract():
    instruction = """
        Generate a simplified intermediate form of the original
    question based on the previous sub-questions or thoughts step by
    step.

        The previous sub-questions or thoughts with marked
    dependencies actually form a directed acyclic graph (DAG), where
    nodes whose dependencies is empty list can be regarded as
    independent sub-questions or thoughts.

        The simplified question must be:
        1. self-contained: The simplified question's description must
    contain all information needed to solve itself, without requiring
    additional information from the original question or reasoning
    trajectory
        2. test-time reduced: The simplified question must require
    fewer reasoning steps compared to the original question (these
    steps are reduced because these solved independent sub-problems or
     thoughts become known conditions in the simplified question or
    excluded as incorrect explorations)

    """
    formatter = "Last step, enclose the question within <question></
    question> tags"
    instruction += formatter
    return instruction

def judge(question: str, solutions: list):
    instruction = """
        Here is the original problem:
        {question}

        Here are some reference solutions:
        {solutions}

        Ensemble the best answer to the original problem from the
    solutions step by step:
    """
    formatter = "Last step, enclose the answer within <answer></answer
    > tags (must be an integer or decimal number without units and
    explanations)"
    instruction += formatter

    solutions_str = ""
    for i, solution in enumerate(solutions):
        solutions_str += f"solution {i}: {solution}\n"
    prompt = instruction.format(question=question, solutions=
    solutions_str)
    return prompt
```

Listing 1: Math

### A.1.2 Code Generation

```
def direct(question: str, contexts: str):
```

```python
    instruction = """
        Solve the following problem step by step:
        {question}
        Your code should be a python function with format: {contexts}

        Please extend your reasoning process as much as possible; the
    longer the chain of thought, the better.

    """
    formatter = "Last step, enclose your code within '''python and '''
    "
    instruction += formatter
    prompt = instruction.format(question=question, contexts=contexts)
    return prompt

def decompose():
    instruction = """
        Decompose the previous reasoning trajectory into a series of
    sub-questions or thoughts.

        Instructions:
        1. Each sub-question or thought should list its other sub-
    questions or thoughts' indexes it depends (0-based, can be an
    empty list)
        2. Dependencies are defined as information needed in sub-
    question or thought that:
            - Does NOT come directly from the original question
            - MUST come from previous sub-questions or thoughts
    """
    return instruction

def contract(dag, test_cases):
    instruction = """
        Generate a simplified intermediate form of the original
    problem based on the variable dependency analysis.

        You ast.arg given a directed acyclic graph (DAG) representing
    the dependencies between variables in the original code:
        {dag}

        And the original test cases:
        {test_cases}

        The simplified problem must be:
        1. Self-contained: The description must contain all
    information needed to solve itself, without requiring additional
    context from the original problem
        2. Test-time reduced: The simplified problem must require
    fewer reasoning steps by using intermediate variables from the
    original code as direct inputs

        Your task is to:
        1. Create a simplified version of the problem that starts with
     intermediate variables as inputs
        2. Generate new test cases that use these intermediate
    variables as parameters while maintaining the exact same expected
    outputs as in the original test cases

        Do not use any code examples in your simplified problem
    formulation.
    """
    formatter = r"Enclose the simplified problem within <question></
    question> tag and the new test cases (assert codes, use \n to
    split each case) within <test></test> tag"
    instruction += formatter
```

```
    prompt = instruction.format(dag=dag, test_cases=test_cases)
    return prompt

def judge(question: str, solutions: list):
    instruction = """
        Here is the original problem:
        {question}

        Here are some reference solutions:
        {solutions}

        Give the index of the best solution as your answer.
    """
    formatter = "Last step, enclose the answer within <answer></answer
    > tags (0-based)"
    instruction += formatter

    solutions_str = ""
    for i, solution in enumerate(solutions):
        solutions_str += f"solution {i}: {solution}\n"
    prompt = instruction.format(question=question, solutions=
    solutions_str)
    return prompt
```

Listing 2: Code

### A.1.3 Multi-hop Question Answering

```
def direct(question: str, contexts: str):
    instruction = """
        Solve the following multi-hop question step by step:
        {question}

        CONTEXTS:
        {contexts}

        Firstly, you need to extract the relevant supporting sentences
     from the original text, then cut out the continuous segments as
    the answer.
    """
        formatter = """
    Provide your response in this JSON format:
    {{
        "question": {question},
        "thought": "give your step by step thought process here",
        "supporting_sentences": [
            "Include ALL sentences needed to justify your answer",
            "Use ... for long sentences when appropriate"
        ],
        "answer": "Your precise answer following the instructions
    above" or "none" if no answer can be found
    }}
    """
    instruction += formatter
    prompt = instruction.format(question=question, contexts=contexts)
    return prompt

def decompose(question: str, trajectory: str, answer: str):
    instruction = """
        You are tasked with breaking down a multiple choice question
    reasoning process into sub-questions.

        Original Question: {question}
```

```
        Complete Reasoning Process: {trajectory}

        Instructions:
        1. Break down the reasoning process into a series of sub-
    questions
        2. Each sub-question should:
            - Be written in interrogative form
            - Have a clear answer
            - List its other sub-questions' indexes it depends (0-based
    , can be an empty list)
        3. Dependencies are defined as information needed to answer
    the current sub-question that:
            - Does NOT come directly from the original question
            - MUST come from the answers of previous sub-questions
    """
    formatter = """
        Format your response as the following JSON object:
        {{
            "thought": "<the thought process of how to step by step
    propose the sub-questions until the answer of the original
    question in the given reasoning process is obtained>",
            "sub-questions": [
                {{
                    "description": "<the description of the sub-
    question>",
                    "answer": <the answer to the sub-question>,
                    "depend": [<indices of the dependent sub-questions
    >, ...]
                }}
            ],
            "answer": "{answer}"
        }}
    """
    return (instruction + formatter).format(question=question,
    trajectory=trajectory, answer=answer)

def contract(question: str, decompose_result: dict, independent: list,
     dependent: list):
    instruction = """
        You are a multiple choice question solver specializing in
    optimizing step-by-step reasoning processes. Your task is to
    optimize the existing reasoning trajectory into a more efficient,
    single self-contained question.

        For the original question: {question}

        Here are step-by-step reasoning process:
        {response}

        {sub_questions}

        Here are explanations of key concepts:
        1. self-contained: The optimized question must be solvable
    independently, without relying on any external information
        2. efficient: The optimized question must be simpler than the
    original, requiring fewer reasoning steps and having a clearer
    reasoning process (these steps are reduced because some solved sub
    -problems become known conditions in the optimized question or are
     excluded as incorrect explorations)

        Note: Since this is a multiple choice question, the optimized
    question must completely retain the options of the original
    question.
```

```
        You can freely reason in your response, but please enclose the
    your optimized question within <question></question> tags
    """
    sub_questions = """
        The following sub-questions and their answers can serve as
    known conditions:
        {independent}

        The descriptions of the following questions can be used to
    form the description of the optimized problem:
        {dependent}

        """
    answer = decompose_result["answer"]
    for sub_q in independent:
        sub_q.pop("depend", None)
    for sub_q in dependent:
        sub_q.pop("depend", None)

    sub_questions = sub_questions.format(independent=independent,
    dependent=dependent)
    return instruction.format(question=question, answer=answer,
    response=decompose_result["response"], sub_questions=sub_questions
    )

def judge(question: str, solutions: list):
    instruction = """
        You are a precise multiple choice question solver. Compare
    then synthesize the best answer from multiple solutions to select
    the most correct option:

        QUESTION: {question}

        SOLUTIONS:
        {solutions}

        Extend your chain of thought as much as possible; the longer
    the chain of thought, the better.

        You can freely reason in your response, even propose new
    reasoning to get a better answer than all solutions, but please
    mark the final option with <answer>single letter of your chosen
    option</answer> tags
    """

    solutions_str = ""
    for i, solution in enumerate(solutions):
        solutions_str += f"solution {i}: {solution}\n"
    prompt = instruction.format(question=question, solutions=
    solutions_str)
    return prompt
```

Listing 3: Multi-hop QA

## A.2 Hyperparameter Configuration

**Maximum Transition Count.**   The maximum number of transitions in the Markovian reasoning chain is a key hyperparameter that controls the depth of reasoning exploration. Theoretically, longer chains enable deeper reasoning, but practical considerations require balancing performance gains with computational efficiency. Throughout our experiments, we uniformly set the maximum transition count to 3, which empirically provides an effective trade-off (see Section B.3 for empirical justification based on structural depth analysis).

**Adaptive Setting.**  For query-specific optimization, the maximum transition count can be dynamically determined by analyzing the initial DAG structure. Since each transition ideally eliminates one layer of independent nodes (those without incoming edges), the depth of the initially decomposed DAG $\mathcal{G}_0$ serves as a reasonable upper bound estimate for the required number of transitions. This can be computed via a simple graph traversal without additional LLM invocations.

**Other Hyperparameters.**  We use temperature $T = 1.0$ for all LLM sampling operations to balance exploration and determinism. For integration experiments with tree-based methods (Section 4.4), we use 3 branches for ToT and vary the number of trees in FoT as {2, 4, 8} to study scaling behavior.

### A.3  Baseline Implementation Details

This subsection describes our implementation of baseline methods to ensure fair and reproducible comparisons.

#### A.3.1  Forest of Thoughts (FoT)

In our implementation, we utilize the classical Tree of Thoughts (ToT) approach as the fundamental tree structure within the Forest of Thoughts framework, while maintaining several critical mechanisms from the original FoT design, including majority voting for aggregating results across different trees and expert evaluation for assessing solution quality.

However, our implementation differs from the original FoT in certain aspects to accommodate a broader range of question types. Specifically, we remove the early stopping criteria that terminate tree splitting when nodes cannot produce valid outputs. While this mechanism is particularly effective for constrained tasks like Game-of-24 where rule-based validation is straightforward, it is less applicable to our diverse evaluation scenarios where output validity is less clearly defined. Instead, we maintain tree expansion regardless of intermediate output quality, allowing the framework to explore potentially valuable paths that might initially appear suboptimal. Additionally, we omit the Input Data Augmentation technique, as analogical reasoning approaches do not demonstrate consistent effectiveness across different question domains in our experiments.

These modifications preserve the core strengths of FoT while enhancing its adaptability to a wider range of reasoning tasks. Our implementation successfully reproduces the scaling curves reported in the original FoT paper and achieves superior performance across multiple benchmarks.

#### A.3.2  AFlow

For AFlow, we adopt the optimal workflows identified in the original work for each benchmark dataset while making necessary adaptations to our experimental setup. For mathematical reasoning tasks on MATH and GSM8K, we directly employ AFlow's proven optimal workflows. For multi-hop reasoning scenarios in LongBench, we use the workflow initially optimized for HotpotQA, as both datasets share core multi-hop reasoning characteristics. This approach ensures we leverage AFlow's strengths while maintaining consistency across similar problem types.

#### A.3.3  Dataset-Specific Details

For the MATH dataset, we filter out questions with non-integer or non-decimal answers to ensure consistent evaluation. We evaluate the first 1,000 cases from MATH for efficiency, while assessing the remaining benchmarks in their entirety.

## B  Empirical Analysis and Validation

This section presents detailed empirical analyses that validate the effectiveness of our framework, including quality metrics for DAG generation, concrete examples of the decomposition-contraction process, and statistical analyses of structural properties.

## B.1 DAG Generation Quality Assessment

To evaluate the quality of our two-phase transition mechanism (decomposition and contraction), we provide comprehensive quality metrics across multiple datasets. Table 2 presents three key metrics that assess different aspects of the DAG generation and state transition process.

Table 2: DAG Generation Quality Metrics Across Benchmarks

| Metric | MATH | GSM8K | MBPP | LongBench |
|---|---|---|---|---|
| Answer Equivalence Maintenance | 99.2% | 99.5% | 99.7% | 99.3% |
| Test-time Complexity Reduction | 76.4% | 82.1% | 74.8% | 79.2% |
| LLM-as-a-Judge Selection Rate | 92.5% | 95.8% | 83.1% | 91.5% |

**Evaluation Methodology.** Both answer equivalence and test-time complexity reduction are assessed through LLM evaluation, where the evaluator LLM is provided with $Q_i$ and $Q_{i+1}$ along with their execution processes. The LLM judges answer equivalence by examining whether the reasoning trajectory's derivation goals remain consistent, and assesses complexity reduction by analyzing the trajectory length and required reasoning steps.

**Metric Definitions.**

- **Answer Equivalence Maintenance**: The probability that the contracted question $Q_{i+1}$ maintains answer equivalence with the original question $Q_0$. The consistently high rates (>99% across all datasets) demonstrate the reliability of our transition mechanism.
- **Test-time Complexity Reduction**: The probability that $Q_{i+1}$ successfully reduces reasoning complexity compared to $Q_i$. Rates between 74%-82% indicate that most transitions achieve meaningful simplification.
- **LLM-as-a-Judge Selection Rate**: The probability that $Q_{i+1}$ is selected (or at least not rejected) by the LLM-as-a-judge when choosing among $\{\text{solve}(Q_i), \text{solve}(\mathcal{G}_i), \text{solve}(Q_{i+1})\}$. High selection rates (83%-96%) confirm that the generated states provide valuable reasoning improvements.

These metrics collectively demonstrate that our DAG generation and contraction process reliably maintains answer equivalence while achieving meaningful complexity reduction, validating the effectiveness of our two-phase transition design.

## B.2 Detailed DAG Decomposition and Contraction Examples

To provide concrete understanding of how the DAG decomposition and contraction mechanism works in practice, we present detailed step-by-step examples demonstrating the complete two-phase transition process.

### B.2.1 Mathematical Reasoning Example

**Original Problem** ($Q_0$): *There are two possible triangles ABC satisfying AB = 10, AC = b > 10, and sin B = 3/5. Find the positive difference between the lengths of side BC.*

**Step 1: Direct Reasoning Trajectory.** When solving $Q_0$ directly, the LLM generates the following reasoning chain:

1. Given sin B = 3/5, we can find cos B using the Pythagorean identity: $\cos^2 B + \sin^2 B = 1$
2. This gives us $\cos^2 B = 1 - (3/5)^2 = 1 - 9/25 = 16/25$
3. Therefore, cos B = ±4/5 (two possible values)
4. Using the Law of Cosines: $b^2 = a^2 + c^2 - 2ac\cos B$, where $a = BC, c = AB = 10$
5. For cos B = 4/5: $b^2 = a^2 + 100 - 20a(4/5) = a^2 - 16a + 100$
6. For cos B = -4/5: $b^2 = a^2 + 100 - 20a(-4/5) = a^2 + 16a + 100$

7. Solving these two cases yields two possible values for BC

8. The positive difference is computed from these two values

**Step 2: DAG Decomposition ($Q_0 \rightarrow \mathcal{G}_0$).** The LLM decomposes this reasoning trajectory into a dependency structure:

**Node 0:** "Calculate cos B from sin B = 3/5 using the Pythagorean identity"

- Dependencies: [] (no dependencies, independent subproblem)
- Result: cos B = ±4/5

**Node 1:** "Given AB = 10, AC = b > 10, and cos B = ±4/5, apply the Law of Cosines to find the two possible values of BC"

- Dependencies: [0] (depends on the result of Node 0)

**Node 2:** "Calculate the positive difference between the two values of BC"

- Dependencies: [1] (depends on the result of Node 1)

The DAG structure is: Node 0 → Node 1 → Node 2, forming a linear chain of depth 3.

**Step 3: Contraction ($\mathcal{G}_0 \rightarrow Q_1$).** Nodes without incoming edges (Node 0) represent independent subproblems that can be directly solved. After solving Node 0, we obtain cos B = ±4/5. This information is incorporated into the problem statement, and nodes depending on it are reformulated:

**Contracted Question ($Q_1$):** *Given that cos B can be either 4/5 or -4/5, with AB = 10 and AC = b > 10, use the Law of Cosines to find the two possible values of BC, then calculate their positive difference.*

**Key observations:**

- $Q_1$ is self-contained: All necessary information (cos B values) is now explicitly stated
- $Q_1$ maintains answer equivalence with $Q_0$: Solving $Q_1$ yields the same final answer
- $Q_1$ has reduced test-time complexity: The trigonometric calculation is eliminated, reducing reasoning steps from 8 to approximately 5
- The DAG depth is reduced from 3 to 2 (only Nodes 1 and 2 remain)

**Step 4: LLM-as-a-Judge Selection.** After generating the triplet $\{\text{solve}(Q_0), \text{solve}(\mathcal{G}_0), \text{solve}(Q_1)\}$, the LLM-as-a-judge evaluates which provides the best answer to the original problem $Q_0$. In this case:

- solve($Q_0$): Direct solution with full reasoning chain
- solve($\mathcal{G}_0$): Solution by explicitly solving each node in the DAG
- solve($Q_1$): Solution of the contracted problem

If $Q_1$ maintains answer equivalence (which it does), solve($Q_1$) will provide a valid answer and is likely to be selected due to its cleaner reasoning structure. If the contraction process had failed to maintain equivalence, solve($Q_1$) would produce an incorrect or nonsensical answer, and the judge would select one of the other options, naturally filtering out the failed transition.

**Iteration Potential.** If we continue from $Q_1$, a second transition could further decompose and contract the problem, potentially separating the two Law of Cosines calculations from the difference computation. This iterative process continues until reaching an atomic state where no further meaningful decomposition is possible.

### B.2.2   Key Insights from the Example

This example illustrates several important aspects of our framework:

1. **Structural Guidance:** The DAG explicitly captures dependencies, allowing the contraction phase to identify which information can be "baked into" the problem statement (Node 0's result) versus which must remain as reasoning steps (Nodes 1-2).

2. **Answer Equivalence:** The contracted question $Q_1$ asks for exactly the same final answer as $Q_0$, ensuring the Markov property holds while making meaningful progress.

3. **Complexity Reduction:** By solving independent subproblems and incorporating their results, $Q_1$ requires fewer reasoning steps, reducing the test-time computational burden.

4. **Implicit Quality Control:** The LLM-as-a-judge mechanism naturally filters failed transitions—if contraction produces an invalid or non-equivalent question, it won't be selected, preventing error propagation.

## B.3 Analysis of Structural Diversity

To understand the structural characteristics of problems decomposed by our framework and provide empirical justification for our hyperparameter choices, we analyze the DAG structures generated from the first 1,000 questions of the MATH dataset.

### B.3.1 Graph Structure and Chain Length

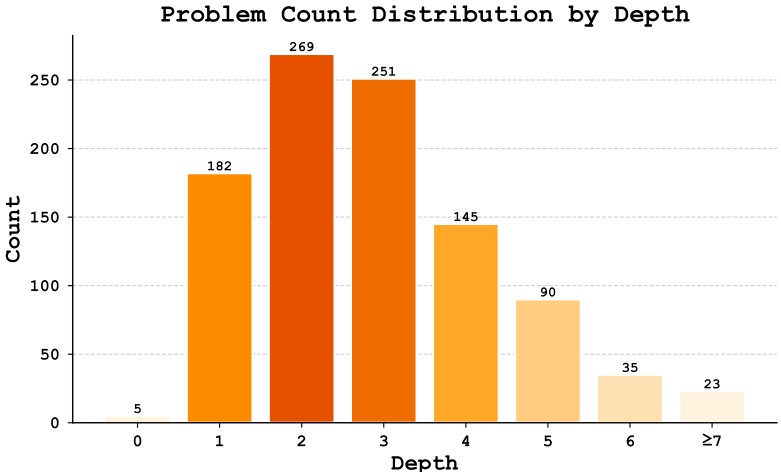

Figure 5: Distribution of solution depths across questions. Darker orange bars indicate depths that appear more frequently in the dataset.

Figures 5 and 6 reveal clear structural patterns in the decomposed questions. The depth distribution (Figure 5) shows that most questions exhibit depths between 2 and 4, with depth 3 being the most common pattern. This observation provides empirical justification for our choice of maximum transition count (3) in the main experiments—the structural depth naturally aligns with the transition requirements for most problems.

Similarly, the subquestion count distribution (Figure 6) indicates that questions typically decompose into 2 to 5 subquestions, with 3-4 subquestions representing the most frequent pattern. These statistics suggest that most reasoning problems naturally decompose into a small number of manageable subproblems, supporting our framework's design assumption that complex reasoning can be effectively simplified through structured decomposition.

### B.3.2 Correlation Between Structural Complexity and Performance

Notably, we observed correlations between these structural metrics and solution accuracy. The scatter plots reveal two important patterns: First, as shown in Figure 8, as the depth of the solution graph increases, there is a general trend of decreasing accuracy. Second, as illustrated in Figure 7, questions with more subquestions tend to show lower accuracy rates. The color intensity of the points provides additional insight - darker points represent more common structural patterns in our dataset, showing

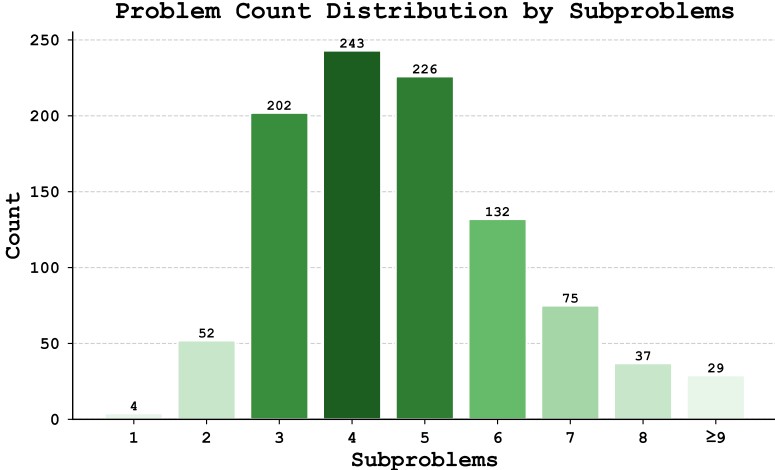

Figure 6: Distribution of subquestion counts across questions. Darker green bars represent more common subquestion counts in the solutions.

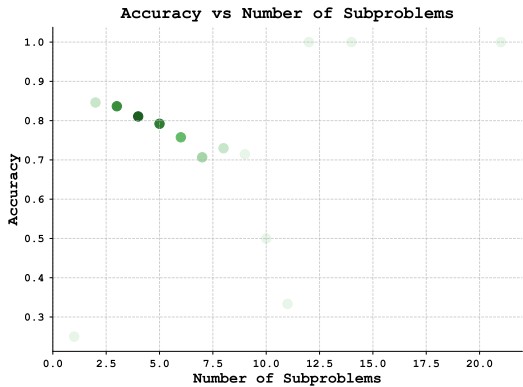

Figure 7: Number of subquestions vs accuracy. Color intensity (green) reflects data density - darker points represent more frequent patterns.

that most of our high-accuracy solutions come from questions with moderate depth and subquestion counts. This suggests that more complex question structures, characterized by either greater depth or more subquestions, pose greater challenges for question-solving systems. The decline in accuracy could be attributed to error propagation through longer solution chains and the increased cognitive load required to maintain consistency across more complex question structures.


Figure 8: Solution depth vs accuracy. Color intensity (orange) reflects data density - darker points represent more frequent patterns.

• Please provide a short (1–2 sentence) justification right after your answer (even for NA).

**The checklist answers are an integral part of your paper submission.** They are visible to the reviewers, area chairs, senior area chairs, and ethics reviewers. You will be asked to also include it (after eventual revisions) with the final version of your paper, and its final version will be published with the paper.

The reviewers of your paper will be asked to use the checklist as one of the factors in their evaluation. While "[Yes] " is generally preferable to "[No] ", it is perfectly acceptable to answer "[No] " provided a proper justification is given (e.g., "error bars are not reported because it would be too computationally expensive" or "we were unable to find the license for the dataset we used"). In general, answering "[No] " or "[NA] " is not grounds for rejection. While the questions are phrased in a binary way, we acknowledge that the true answer is often more nuanced, so please just use your best judgment and write a justification to elaborate. All supporting evidence can appear either in the main paper or the supplemental material, provided in appendix. If you answer [Yes] to a question, in the justification please point to the section(s) where related material for the question can be found.

IMPORTANT, please:

• **Delete this instruction block, but keep the section heading "NeurIPS Paper Checklist",**
• **Keep the checklist subsection headings, questions/answers and guidelines below.**
• **Do not modify the questions and only use the provided macros for your answers**.

