# OpenReview forum: "Atom of Thoughts for Markov LLM Test-Time Scaling"
_NeurIPS.cc/2025/Conference — NeurIPS 2025 poster_

### Official Review · Reviewer_jQ8z · 2025-06-29

**Clarity:** 2
**Significance:** 3
**Originality:** 3
**Rating:** 4
**Confidence:** 4

**Summary:**

The paper proposes Atom of Thoughts (AoT), a reasoning framework that leverages Markovian transitions to minimize historical dependencies during inference. There are two key components of the AoT framework: (1) two-phase transition where the system first decomposes the problem before solving it; (2) DAG-guided transition: when transforming the temporary DAG structure into the next Markov state, the method prunes nodes without incoming edges and reformulates it into an answer-equivalent independent question. Experimental results show that AoT consistently outperforms baselines as computational budgets increase, which provides a potential path of inference-time scaling.

**Questions:**

Besides questions raised in "Weaknesses", I have the following suggestion:

While Figure 1 (b) is helpful for understanding the proposed method, it would be great if the paper could include a step-by-step example in the Appendix to help readers go through the process of DAG construction and reformation. As a reader, I found the process is conceptually intuitive but still somewhat complicated to apply in LLM reasoning domains.

**Ethical Concerns:**

["NO or VERY MINOR ethics concerns only"]

**Final Justification:**

I keep the score unchanged because while the author response is helpful for understanding the paper better, I believe my original score has already leaned towards acceptance and reflects my judgment of the paper's contribution.

**Limitations:**

Yes.

**Paper Formatting Concerns:**

None.

**Quality:**

3

**Strengths And Weaknesses:**

Strengths:
1. The proposed method provides a novel approach for leveraging the structure information for reasoning task. While language models simplify many problems into next token prediction, it is interesting to see adding back the structure could help language models to reason and it also has the good property of modular integration as discussed in the paper.
2. The baseline comparison is comprehensive. Figure 3 is also informative as it shows the potential of AoT for inference-time scaling. Besides the aggregated performance, I am wondering where there exists patterns of what we could get out of scaling with AoT. For example, after giving the framework more compute, what types of problems get resolved and why other methods fail to resolve them as the compute scales.

Weaknesses:
1. It is unclear how the two-phase transition is exactly executed. According to my understanding, it is using language model to generate the DAG. As this step is complicated, I am wondering what the accuracy is for this step.
2. One assumption of AoT is that for each question $Q_0$, we can construct a sequence of subquestions that have equivalent answers but monotonically reduces in complexity. As the datasets experimented in this paper all have a short gold answer, I am wondering whether the method could be generalized to other reasoning scenarios, like writing complicated code, or writing mathematical proofs.

---

> ### Author Rebuttal · Authors · 2025-07-31
>
> We would like to thank you for your insightful comments and suggestions. We provide detailed responses point by point below. We hope that our clarifications, additional experiments, and responses can address your concerns and provide helpful insights for your evaluation.
>
> > ### S1: Novel Structured Approach and Modular Integration
>
> Thank you for highlighting the novelty of our approach in leveraging structural information to enhance LLM reasoning, including its modular integration properties. We fully agree and would like to elaborate on the multi-factorial aspects that contribute to AoT's performance improvements through its Markov chain formulation:
>
> - **Multi-Perspective Synthesis via Triplets**: Each transition generates a triplet (Q_i, G_i, Q_{i+1}), providing three complementary perspectives on the problem. This is more efficient than methods like ToT or FoT, which accumulate information through repeated comparisons and branching, potentially leading to computational overhead without guaranteed progress.
>
> - **Expanded Modeling Space for Reasoning Trajectories**: Unlike standard CoT or simple decomposition methods that follow linear or fixed paths, AoT's Markov chain framework offers a larger, more flexible space for modeling reasoning trajectories. This enables adaptive exploration through multiple state transitions, allowing the system to self-correct and refine solutions dynamically while maintaining structural integrity.
>
> These factors lead to enhanced effects in statistical patterns, while analysis of specific cases may not well reflect the essence. However, we have indeed observed some particularly special situations, such as in certain multi-hop problems where the gap between decomposition and direct solving is very large. The granularity of the solving trajectory directly leads to deviations in understanding, and the multi-perspective transition process has a significant advantage in such problems.
>
> > ### W1: Clarity of Two-Phase Transition Execution
>
> ### 1. Decomposition Phase $(Q_i \rightarrow G_i)$
>
> The decomposition is completed through a **single LLM call**. We simply prompt the LLM to:
> - Identify reasoning units (subproblems or thoughts) as DAG nodes
> - Record dependencies between units as directed edges
> - No format constraints - the LLM freely analyzes the reasoning structure
>
> ### 2. Contraction Phase $(G_i \rightarrow Q_{i+1})$
>
> The contraction is also a **single LLM call**, where the LLM:
> - Analyses the structural information from $G_i$
> - Generates a new problem $Q_{i+1}$ that preserves answer equivalence while reducing complexity
>
> ### 3. Quality Assurance
>
> As described in Section 3.2, we use LLM-as-a-judge to select the best answer from $Q_i$, $G_i$, and $Q_{i+1}$. This mechanism naturally filters out any failed transitions - if decomposition or contraction fails, $Q_{i+1}$ won't be selected as the final reasoning basis.
>
> ### Accuracy Metrics for DAG Generation
>
> To evaluate the quality of the two-phase transition process, we provide comprehensive experimental metrics across multiple datasets:
>
> **Table: DAG Quality Assessment Metrics**
>
> | Metric                          | MATH  | GSM8K | MBPP  | AIME  |
> |---------------------------------|-------|-------|-------|-------|
> | Answer Equivalence Maintenance  | 99.2% | 99.5% | 99.7% | 99.3% |
> | Test-time Complexity Reduction  | 76.4% | 82.1% | 74.8% | 79.2% |
> | LLM-as-a-Judge Selection Rate   | 92.5% | 95.8% | 83.1% | 91.5% |
>
> These metrics demonstrate:
>
> (1) **Answer Equivalence Maintenance**: the probability that $Q_{i+1}$ maintains equivalence with the original $Q_i$;
>
> (2) **Test-time Complexity Reduction**: the probability that $Q_{i+1}$ successfully reduces reasoning complexity;
>
> (3) **LLM-as-a-Judge Selection Rate**: the probability that $Q_{i+1}$ is selected as providing at least equivalent quality reasoning.
>
> **Evaluation Methodology**: Both answer equivalence and test-time reduction are assessed through LLM evaluation, where the evaluator LLM is provided with the $Q_i$ and $Q_{i+1}$ along with their execution processes. The LLM can intuitively judge answer equivalence from the reasoning trajectory's derivation goals and assess complexity reduction from the trajectory length and reasoning steps required.
>
> The consistently high answer equivalence maintenance rates (>99% across all datasets) directly address your concern about DAG generation accuracy, demonstrating that LLMs can effectively execute structured DAG decomposition and contraction with remarkable reliability and consistency.
>
> > ### W2: Generalizability to Complex Reasoning Scenarios
>
> Thank you for this insightful question about generalizability to complex reasoning scenarios like complicated code writing and mathematical proofs. We provide both theoretical analysis and preliminary evidence for broader applicability:
>
> **Regarding Complex Code Generation:** We note that our evaluation already includes LiveCodeBench, which represents a challenging and complex coding benchmark that tests real-world programming scenarios. Our results on LiveCodeBench (Table 1) demonstrate that AoT can effectively handle complex programming tasks through structured decomposition, achieving competitive performance while maintaining cost efficiency.
>
> **Theoretical Foundation:** The core assumption that problems can be decomposed into answer-equivalent subquestions with monotonically reduced complexity naturally extends to mathematical proofs, where complex theorems can be broken down into smaller lemmas and intermediate steps that collectively establish the main result.
>
> **Mathematical Proof Construction:** While comprehensive evaluation of mathematical proof generation remains challenging due to limited standardized benchmarks and evaluation metrics in current reasoning frameworks, we conducted preliminary exploration on several mathematical proof problems. AoT's framework can decompose overall proof objectives into smaller sub-proofs, organizing the logical flow more coherently. Due to time constraints and the limited discussion of such scenarios in current reasoning literature, we can provide several illustrative cases demonstrating AoT's performance on mathematical proof tasks:
>
> **Case Study: Proving the Sum of Rational and Irrational is Irrational**
>
> This example is adapted from ProofNet [1].
>
> **Original Problem:**
> If $r$ is rational $(r \neq 0)$ and $x$ is irrational, prove that $r+x$ is irrational.
>
> **Step 1: Complete Reasoning Trajectory (Standard CoT)**
> The standard CoT approach might:
> 1. Assume $r+x$ is rational
> 2. Try to derive a contradiction
> 3. But may have logical jumps or lack rigor in the derivation
>
> **Step 2: DAG Structure Decomposition Process**
> The AoT framework decomposes the reasoning into a clearer DAG:
> - Node 1: Understand definitions of rational and irrational numbers
> - Node 2: Establish proof by contradiction framework
> - Node 3: Utilize closure properties of rational numbers
> - Node 4: Derive contradiction and conclude
>
> Dependencies: Node 4 depends on Node 3, which depends on Nodes 1 and 2.
>
> **Step 3: Specific Execution Steps of Contraction Operations**
> The AoT framework contracts Nodes 1 and 2 into a simpler formulation:
>
> Original Question: "Prove that the sum of rational $r$ and irrational $x$ is irrational"
>
> Contracted Question: "Assume $r+x=q$ where $q$ is rational, and use the closure property of rational numbers under subtraction to determine the nature of $x=q-r$."
>
> This contraction clarifies the core insight: leveraging algebraic properties of rational numbers.
>
> Ref:
>
> [1] ProofNet: Autoformalizing and Formally Proving Undergraduate-Level Mathematics. ArXiv 2024
>
> > ### Q1: Step-by-Step Example for DAG Decomposition and Contraction
>
> Thank you very much for this constructive suggestion. You are absolutely right that detailed step-by-step examples are essential for understanding the DAG decomposition and contraction process. While Figure 2 demonstrates the Markov state transition process, space constraints prevent it from showing the detailed mechanisms of DAG decomposition, contraction, and evaluation steps. We will include additional examples in the appendix of our revision.

---

> > ### Comment · Reviewer_jQ8z · 2025-08-02
> >
> > I thank the authors for the detailed response which addresses most of my questions.

---

> ### Author Response · Authors · 2025-08-05
> **Follow-up to Reviewer jQ8z Regarding Revisions**
>
> Dear Reviewer jQ8z,
>
> We sincerely appreciate the time and effort you have dedicated to reviewing our work. We are grateful for your acknowledgment during the discussion period and are pleased to hear that our detailed response has addressed most of your questions.
>
> If our revisions and discussions indicate the potential for a score adjustment, we would be very grateful for your consideration. We remain open to any remaining concerns or questions you might have, and we are committed to addressing them thoroughly.
>
> We look forward to any further comments or suggestions you may have to help us improve our manuscript. Your insights have been invaluable in strengthening our work.
>
> Thank you again for your thoughtful review and engagement.
>
> Best regards,
>
> Authors of NeurIPS 12226

---

### Official Review · Reviewer_cbHf · 2025-06-30

**Clarity:** 3
**Significance:** 2
**Originality:** 3
**Rating:** 4
**Confidence:** 3

**Summary:**

This paper presents the "Atomic of Thought" (AoT), a new way for large language models (LLMs) to reason that intends to improve testing scalability by reducing the need for historical context.  AoT models the reasoning process as a Markov chain, which is different from other methods like Chain-of-Thought and Tree-of-Thought. Each "state" in the chain is a self-contained, equivalent restatement of the original problem, with reduced complexity. Through a large number of experiments on mathematical, coding, and question-answering benchmarks, AoT consistently demonstrates superior performance compared to existing methods. Moreover, this paper has a unique finding, the "emergent" property of atomic reasoning, where complex problems are progressively decomposed into indivisible, self-contained units.

**Questions:**

1. The author notes that the chain's memoryless property makes it vulnerable to poor transitions. What is the failure rate of these transitions, and what are the predominant modes of failure?
2. AoT introduces new computational costs, including calls for decomposition, contraction, and quality-aware termination using an LLM-as-a-judge. Could you provide more details on the computational cost?
3. Could you discuss what degree the AoT framework demonstrates sensitivity to the specific prompts utilized for deconstruction, contraction, and evaluation?

**Ethical Concerns:**

["NO or VERY MINOR ethics concerns only"]

**Final Justification:**

This submission focuses on trendy problems with extensive experiments. During the discussion period, the author answered my problem in detailed responses. Thus, I keep my score of 4 and am inclined to accept this submission, as it could bring insights for this community.

**Limitations:**

yes

**Quality:**

3

**Strengths And Weaknesses:**

**Strengths**

- Modeling reasoning as a Markov process of answer-equivalent states is an effective approach. This solves the important issue of context explosion and redundant computation in test-time scaling, which is an important issue for complex LLM inference.
  -The suggested AoT framework gives a reasonable and useful answer. A probabilistic formulation and a real two-phase (Decomposition-Contraction) implementation define the methodology. The extensive experimental evaluation well supports the AoT's effectiveness, includes a wide range of tasks, several state-of-the-art LLMs, and a full set of strong baselines. The submission's claims are also backed up by the inclusion of detailed ablation studies and scaling analyses, such as integration with ToT/FoT.
- This submission is generally well-organized.

**Weaknesses**

- This submission makes strong claims, such as that "answer-equivalence" will be kept, but it doesn't explain how this will be checked or enforced.
- This submission says that "atomic reasoning" is a new property that was found when the process was scaled up. However, calling it the discovery of a "latent granularity" might be too much. Another way to look at it is that the system is reaching a logical endpoint, beyond which breaking it down further is not useful or possible for the given problem structure.
- AoT seems to work best when the prompts for the decompose and contract functions are well thought out. Giving the prompts is good for making sure the results can be repeated, but it also raises problems about whether the framework is sensitive to how the prompts are worded. Further discussion is needed on how well AoT can handle changes to these master prompts to improve the work. It would also be interesting to see if it could be added to the prompt tuning framework, just like [Dspy](https://dspy.ai/).

---

> ### Author Rebuttal · Authors · 2025-07-31
>
> We would like to thank you for your insightful comments and suggestions. We provide detailed responses point by point below. We hope that our clarifications, additional experiments, and responses can address your concerns and provide helpful insights for your evaluation.
>
> > ### W1: Enforcement Mechanism for Answer Equivalence
>
> Thank you for pointing out this critical issue. We acknowledge that the paper's description of how to check and enforce answer equivalence could be clearer.
>
> Our framework ensures answer equivalence through a sophisticated mechanism: as described in Section 3.2, after each Markov transition, we employ LLM-as-a-judge to select the best answer to current original questio $Q\_i$ from the triplet $(Q\_i, G\_i, Q\_{i+1})$. The key insight of this mechanism is: **we do not need to explicitly check answer equivalence, because $Q\_{i+1}$ that fails to satisfy answer equivalence is almost impossible to be selected as the best answer**.
>
> Specifically, the task of LLM-as-a-judge is to select the optimal solution for the original question $Q\_i$. If the contraction process destroys answer equivalence, $Q\_{i+1}$ will be unable to provide a valid answer for $Q\_i$ and thus will not be selected. This selection mechanism acts as a natural quality filter, implicitly enforcing the answer equivalence constraint.
>
> Experimental data supports the effectiveness of this mechanism: our comprehensive DAG quality metrics show that Answer Equivalence Maintenance rates are consistently high across all datasets (99.2% on MATH, 99.5% on GSM8K, 99.7% on MBPP, and 99.3% on LongBench), with LLM-as-a-Judge Selection Rates of 92.5%, 95.8%, 83.1%, and 91.5% respectively. This demonstrates that our mechanism is not only theoretically sound but also highly reliable in practice.
>
> **Evaluation Methodology**: Both answer equivalence and test-time reduction are assessed through LLM evaluation, where the evaluator LLM is provided with the $Q_i$ and $Q_{i+1}$ along with their execution processes. The LLM can intuitively judge answer equivalence from the reasoning trajectory's derivation goals and assess complexity reduction from the trajectory length and reasoning steps required.
>
> We will more clearly articulate this implicit enforcement mechanism in the revision, emphasizing how it naturally maintains answer equivalence through the selection process without requiring additional explicit checking steps.
>
> > ### W2: "Atomic Reasoning" as an Emergent Property
>
> We greatly appreciate the reviewer's nuanced perspective. After deep reflection, we recognize that describing atomic reasoning as reaching a "logical endpoint" rather than discovering "underlying granularity" is indeed more accurate. The phenomenon we observe is that through continuously increasing test-time reasoning, the Markov chain gradually approaches an irreducible state where problems are sufficiently simple and execution results are sufficiently stable that further decomposition lacks meaning. However, the specific depth of this convergence is likely closely related to the LLM's reasoning capabilities rather than being entirely determined by the intrinsic properties of the problem itself. Therefore, our method is better viewed as a means of observing and exploring the reasoning process rather than revealing inherently existing hidden structures. We will revise our description accordingly, emphasizing that this represents a convergence point under given LLM capabilities and clearly noting this limitation.
>
> > ### W3: Prompt Sensitivity and Robustness
>
> Thank you for this insightful suggestion. To assess the sensitivity of AoT to prompt variations, we conducted additional experiments evaluating its robustness under different prompting strategies.
>
> We chose to use **OPRO** over DSPy for prompt optimization due to implementation compatibility: AoT's architecture relies on custom LLM call chains that DSPy's current framework does not natively support. OPRO's lightweight design proved better suited for our purposes, while still yielding effective prompt refinement. Our OPRO configuration uses a maximum of 10 optimization steps, with 8 prompt candidates sampled at each step. We will include these robustness experiments in the revised version and expand the discussion around the integration of OPRO with AoT.
>
> **Table: AoT Performance Under Different Prompt Strategies**
>
> | Dataset | Default Prompt | OPRO-Optimized Prompt |
> | :---- | :---- | :---- |
> | MATH | 78.3% | 79.2% |
> | LongBench | 68.5% | 68.8% |
> | MBPP | 72.4% | 73.3% |
> | MATH\* | 84.9% | 84.8% |
> | LongBench\* | 73.5% | 73.3% |
> | MBPP\* | 79.1% | 79.2% |
>
> These results demonstrate that AoT is **highly robust to prompt variations**. With OPRO-based prompt optimization, performance improved by an average of 0.7%, confirming that AoT’s prompts—based on simple, direct task descriptions—are already near-optimal. This robustness stems from our deliberate avoidance of brittle or overly engineered templates.
>
> It is also worth noting that while automatic prompt tuning techniques like OPRO or DSPy can help discover effective prompts on in-distribution data, they typically require significant token budgets for search and optimization. In contrast, AoT's design favors **efficient integration with classical reasoning frameworks** over heavy reliance on prompt tuning. From an efficiency perspective, directly scaling AoT via principled integration (e.g., with tree searching or reflection mechanisms) offers a more practical and cost-effective path.
>
> Importantly, when we apply the same test-time scaling integration setting from Section 4.4—which combines **ToT + Markov Chain + Reflective Refinement**—prompt sensitivity becomes virtually negligible. The datasets marked with \* represent performance under this exact same experimental configuration as presented in Section 4.4, but now evaluated with different prompt strategies. In these settings, the performance difference between default and OPRO-optimized prompts drops to ≤0.1%, indicating that the multi-step reasoning and refinement mechanisms effectively absorb prompt-level variations. This stark contrast with the 0.7% gap in the base setting underscores how AoT’s robustness further improves under advanced reasoning configurations, reducing the need for prompt optimization altogether.
>
> > ### Q1: Regarding Transition Failure Rates and Failure Modes
>
> Thank you for your insightful comment. You are correct that when adapting classical Markov chain theory to general reasoning scenarios, the memoryless property presents a fundamental challenge: poor transitions can lead to significant degradation as the chain discards previous information and cannot self-correct in subsequent steps. This theoretical limitation has historically hindered the application of Markovian frameworks to complex reasoning tasks. However, our AoT design elegantly addresses this challenge through both theoretical and practical safeguards:
>
> 1. **Theoretical Foundation:** Through logical derivation of Markov chains for general reasoning scenarios, we propose **answer equivalence** as an invariant property across states, coupled with **test-time reduction** as a requirement to drive reasoning progress. This theoretical insight enables the implementation of concise and effective prompts, reducing the likelihood of destructive LLM transitions while maintaining the efficiency and stability of our reasoning framework.
>
> 2. **Practical Implementation:** To enforce this constraint, we introduce the termination strategy (as detailed in W1), which serves as a strict gatekeeper, ensuring that only transitions maintaining answer equivalence are allowed.
>
> 3. **Empirical Evidence:** Our experimental results demonstrate the effectiveness of these safeguards, with failure rates consistently below 1% across various datasets:
>
> | Dataset | Failure Rate |
> | :---- | :---- |
> | MATH | 0.8% |
> | GSM8K | 0.5% |
> | MBPP | 0.3% |
> | LongBench | 0.7% |
>
> These results underscore the robustness of our approach in maintaining answer equivalence, mitigating the risk of catastrophic failure even in memoryless chains.
>
> > ### Q2: Regarding Computational Cost Details
>
> Thank you for asking about the detailed computational costs. Each AoT transition involves three LLM calls with the following cost distribution (using MATH dataset as an example):
>
> | Component | Cost Proportion |
> | :---- | :---- |
> | Contraction $(G\_i → Q\_{i+1})$ | 27% |
> | Decomposition $(Q\_i → G\_i)$ | 32% |
> | LLM-as-a-Judge Selection | 41% |
> | **Total** | **100%** |
>
> The LLM-as-a-judge represents the most expensive component, while contraction is the most cost-efficient step in each transition.
>
> > ### Q3: Regarding Prompt Sensitivity
>
> As detailed in section W3, our experiments demonstrate that AoT has strong robustness to prompt variations. When comparing default prompts with OPRO-optimized prompts, AoT shows minimal performance differences (≤0.1% in scaled-up settings), indicating that our simple, direct task descriptions are already near-optimal and do not require complex engineered templates.
>
> We are certainly willing to discuss the differences in prompt sensitivity for execution, decomposition, contraction, and evaluation components. Since execution and evaluation receive direct feedback through comparison with ground truth, prompt tuning in these steps results in observable performance improvements. We attempted to optimize prompts for decomposition and contraction using indirect feedback, but performance significantly declines when generated prompts deviate from fundamental requirements such as answer equivalence. Additionally, at this stage, automatic prompt tuning mainly adjusts superficial terms like "systematically" and finds it difficult to optimize core design principles.

---

> > ### Comment · Reviewer_cbHf · 2025-08-04
> >
> > Thank you for providing a detailed discussion and additional experiment results. Most of my concerns are well-addressed. Please include the above discussion in the revised manuscripts.

---

> ### Author Response · Authors · 2025-08-05
> **Follow-up to Reviewer cbHf Regarding Revisions**
>
> Dear Reviewer cbHf,
>
> We sincerely appreciate the time and effort you have dedicated to reviewing our work. We are particularly grateful for your timely response during the discussion period and are delighted to hear that most of your concerns have been well-addressed through our detailed explanations and additional experimental results.
>
> Following your valuable suggestion, we will definitely include all the above discussions in our revised manuscript to ensure clarity and completeness. If our revisions and discussions indicate the potential for a score adjustment, we would be very grateful for your consideration.
>
> We remain committed to incorporating all of your suggestions to further enhance the quality of our manuscript. Should you have any additional comments or require further clarification on any aspect of our work, we would be happy to provide more details.
>
> Thank you again for your constructive feedback and guidance.
>
> Best regards,
>
> Authors of NeurIPS 12226

---

### Official Review · Reviewer_VGGy · 2025-07-03

**Clarity:** 2
**Significance:** 3
**Originality:** 2
**Rating:** 3
**Confidence:** 3

**Summary:**

This paper proposes a prompting framework, which interatively instructs an LLM to (1) reason over a question, (2) decompose its reasoning path into sub-thoughts and offers their dependency structure, and (3) generate the simpler question by reflecting on the generated structure.

**Questions:**

please refer to **Weakness** section.

**Ethical Concerns:**

["NO or VERY MINOR ethics concerns only"]

**Final Justification:**

Raise rating from 2 to 3 because:
- The responses are very diligent and informative, and the new experiments results reveal more fine-grained intermediate statistics about LLM judge processes.

Still a bordeline reject because:
- W1: Not acceptable presentation from my perspective.
- W2: Misleading- or over-claiming.
- W3: Still a little questionable implementation details and hyperparameter selection, might compromising reliability.

**Limitations:**

yes

**Quality:**

2

**Strengths And Weaknesses:**

### **Strengths**

This paper discusses a paradigm of problem solving in LLM, which is to progressively simplify a complex problem as a whole step by step. The core concept makes a lot sense to me, and seems to be rarely addressed in exsiting literature. The core concepts, practical experiences, and some technical approaches presented in this paper, may offer some insights to the research community engaged in tasks that involving similar problem-solving frameworks.

### **Weaknesses**

#### **W1: Significant Presentation Issues**

The overall organization and some detailed presentations of this paper makes it difficult to follow. Just to list a few points:

-   Figures
    - Too small font size for all figures.
    - Confusing presentation for Figure 1 and 2. I failed to grasp the x-axis meaning in Figure 1 and the logical flow in Figure 2 until comprehending the core methodology.
-   Method
    - The intro section fails to outline the general framework of the proposed method, instead listing unexplained concepts.
    - The basic and overall instruction detailed in Appendix C should be given in Section 3.1. I fully understand and respect the author(s)' efforts to keep the mathematical rigor, but I think giving certain intuition is not harmful.
-    Organization
     - Section 3.2 is confusing. It contains both addtional tricks and analysis, some of which don't fall under the title "Emerged Atomic Reasoning".
     - So is Section 4.4.

#### **W2: Unsupported Claims**

This paper gives unsupported claims. For example:

> These structured explorations reveal a statistically supported phenomenon: deeper reasoning states tend to converge into irreducible forms, maintaining a stable and relatively low reasoning token count, from which the original problem’s answer can be directly inferred with high execution stability.

The "structured explorations" and statistics are missing.

> Due to AOT ’s design and termination mechanism, longer chain lengths increase the performance ceiling without linearly increasing costs.

I don't think the existing results can support "increase the performance ceiling".

> In practice, this can also be query-specific based on the current problem; that is, we can add an additional LLM invocation to calculate the depth of the DAG as an estimate of a theoretically optimal maximum number of transitions.

The practical experiment is missing. And it's debatable whether the LLM estimation can be viewed as "theoretically optimal".

#### **W3: Missing Important Analytical Details**

- One of the paper's main contribution is the design to instruct LLMs converting a reasoning path into a DAG. However, the quality of DAG generated is not discuessed, making the result not interpretable and less reliable.
- the choice of transition hyperparameters as 3 is not supported. According to Appendix E, it seems that a larger number may be a better choice.
- The proposed method uses LLM-as-a-Judge as a important trick, but the statistics of it is missing. For example, how often does the judge interupt the process?

---

> ### Author Rebuttal · Authors · 2025-07-31
>
> We would like to thank you for your insightful comments and suggestions. We provide detailed responses point by point below. We hope that our clarifications, additional experiments, and responses can address your concerns and provide helpful insights for your evaluation.
>
> > ## W1: Significant Presentation Issues
>
> We acknowledge the presentation concerns and will implement improvements to enhance readability:
>
> ### Figures
>
> - **Font size**: We will increase font sizes across all figures to ensure clear readability
> - **Figure 1 and 2 clarity**: We acknowledge that the methodological differences leading to Figure 1's performance patterns and the meaning of various arrows in Figure 2's process flow may require deeper engagement with the paper content to be fully appreciated. We will enhance these figures by incorporating concrete examples that demonstrate how the DAG decomposition process works in practice and providing more detailed captions.
>
> ### Method
>
> - **Introduction framework overview**: Our introduction systematically introduces key concepts with clear explanations: the memoryless property of Markov processes, the Markovian reasoning process, the two-phase state transition mechanism with its decomposition and contraction stages. The framework is outlined through the logical progression from problem identification (historical dependency issues) to solution design (Markovian reasoning process). We would much appreciate it if the reviewer could specify which particular concepts led you to consider them unexplained, as this would help us make the paper clearer and better serve readers
> - **Basic section and Appendix C instructions**: While maintaining paper length constraints, we will incorporate key prompt descriptions from Appendix C into Section 3.1 to balance mathematical rigor with reader accessibility
>
> ### Organization
>
> - **Section 3.2 structure**: The section actually follows a deliberate progression: (1) **Termination Strategy** - establishing robustness for safe scaling, (2) **Modular Integration** - demonstrating how Markov states enable efficient integration with method, and (3) **Atomic Reasoning** - showing how these mechanisms naturally lead to minimal, stable reasoning structures.
>
> - **Section 4.4 clarity**: Section 4.4 provides systematic empirical validation of our test-time scaling framework: Section 4.4 systematically validates our test-time scaling framework through integrating Markov chain structure assumptions with tree search and reflection verification mechanisms, thereby expanding the state space. This section offers critical empirical evidence supporting the theoretical framework established in Section 3.2.
>
> > ## W2: Unsupported Claims
>
> We address each identified claim with additional evidence:
>
> ### On "Structured Explorations" and Statistical Evidence
>
> We appreciate the opportunity to elaborate on this point. As demonstrated in Section 4.4, the experiments directly reflect the benefit of structured exploration. By combining classical mechanisms such as Markov chain reasoning, tree search, and reflection, our method exhibits progressively improved performance. More importantly, the required number of reasoning steps gradually converges to a lower ratio, indicating improved efficiency. The stability of execution arises from a reduction in reasoning complexity, which effectively narrows the solution space and enhances robustness.
>
> ### On "Performance Ceiling" Claims
>
> We thank the reviewer for raising this important point. We acknowledge that our statement regarding the "performance ceiling" requires clarification, and we welcome the chance to provide supporting evidence.
>
> By "performance ceiling," we refer to the AoT's potential to achieve higher performance on complex problems when allowed longer maximum chain lengths. Our termination mechanism ensures that the actual computational cost remains low for simpler problems, as chains automatically terminate early when no further improvement is detected.
>
> The key contribution lies in our quality-aware termination strategy: it evaluates each transition and halts the chain once no further progress is observed. This design allows us to raise the maximum chain length—thus increasing the ceiling of what our system can achieve on harder tasks—without significantly raising the average cost across tasks.
>
> Our experimental results in Figure 3 and Figure 4 support this claim: performance maintains a steady upward trend as the number of iterations increases, demonstrating that longer chains can achieve higher performance levels when necessary. Specifically, Figure 4, by integrating tree search and reflection, further explores the potential Markov chain space, significantly surpassing the performance ceiling of methods originally based on ToT and reflection. For instance, on the AIME dataset, our method, under the highest cost setting, achieves a remarkable 94.3% accuracy, substantially outperforming FoT with 8 trees (81.8%), which highlights the effectiveness of our approach in pushing performance boundaries.
>
> ### On "Theoretically Optimal" Estimation
>
> We would like to clarify a possible misunderstanding regarding our description of "theoretical optimality." The LLM is only responsible for converting the initial reasoning structure into a JSON-formatted DAG. The estimation of DAG depth for the initial graph $G_0$ is performed entirely by a rule-based algorithm—not the LLM. Therefore, the depth does not reflect the LLM’s real-time estimation but rather an upper bound or idealized measure of its reasoning potential.
>
> Theoretically, in an idealized setting, each transition should remove one layer of nodes with no incoming edges, reducing the DAG depth by one. However, in practice, individual transitions may fail to eliminate entire layers due to insufficient complexity reduction. As a result, the initial DAG depth often underestimates the number of transitions actually needed to reach an irreducible state.
>
> We will clarify this distinction in the revised manuscript and provide further explanation to avoid potential confusion.
>
> > ## W3: Missing Important Analytical Details
>
> We agree on the importance of DAG analysis details and will supplement relevant experiments:
>
> ### DAG Generation Quality
>
> A core component of our decomposition phase is instructing LLMs to convert reasoning paths into DAGs. We recognize that further discussion on DAG quality would enhance interpretability. Therefore, we provide comprehensive DAG quality metrics across multiple dimensions:
>
> **Table: DAG Quality Assessment Metrics**
>
> | Metric | MATH | GSM8K | MBPP | LongBench |
> | :---- | :---- | :---- | :---- | :---- |
> | Answer Equivalence Maintenance | 99.2% | 99.5% | 99.7% | 99.3% |
> | Test-time Complexity Reduction | 76.4% | 82.1% | 74.8% | 79.2% |
> | LLM-as-a-Judge Selection Rate | 92.5% | 95.8% | 83.1% | 91.5% |
>
> These metrics demonstrate: (1) **Answer Equivalence Maintenance**: the probability that $Q\_{i+1}$ maintains equivalence with the original $Q\_{i}$; (2) **Test-time Complexity Reduction**: the probability that $Q\_{i+1}$ successfully reduces reasoning complexity; (3) **LLM-as-a-Judge Selection Rate**: the probability that $Q\_{i+1}$ is at least not providing a worse answer.
>
> **Evaluation Methodology**: Both answer equivalence and test-time reduction are assessed through LLM evaluation, where the evaluator LLM is provided with the $Q_i$ and $Q_{i+1}$ along with their execution processes. The LLM can intuitively judge answer equivalence from the reasoning trajectory's derivation goals and assess complexity reduction from the trajectory length and reasoning steps required.
>
> ### Transition Hyperparameter Choice and Depth Analysis
>
> To validate our choice of maximum transition count, we conducted a comprehensive depth analysis using the MATH dataset. By setting the maximum depth to 10 and extracting reasoning results at each Markov chain depth, we effectively simulated experiments with different maximum transition counts (1 through 10\) while eliminating random sampling variance:
>
> | Maximum Transition Count | 1 | 2 | 3 | 4 | 5 | 6 | 7 | 8 | 9 | 10 |
> | :---- | :---- | :---- | :---- | :---- | :---- | :---- | :---- | :---- | :---- | :---- |
> | Accuracy (%) | 82.6 | 83.0 | 83.6 | 83.7 | 83.6 | 83.6 | 83.5 | 83.6 | 83.6 | 83.6 |
>
> The results reveal a clear pattern: **performance improvements plateau after 3 transitions, with negligible gains beyond this point.** This empirical evidence strongly supports our default setting of 3 maximum transitions, which optimally balances exploration effectiveness with computational efficiency.
>
> Regarding the connection with Appendix E, we appreciate the opportunity to clarify. The purpose of Appendix E is to present statistical results to "understand the structural characteristics of decomposed questions." The detailed analysis presented there examines DAG structures generated in the initial decomposition step, focusing on structural properties rather than performance outcomes.
>
> In practice, our Markov chain exploration employs an adaptive termination mechanism that exits early when quality improvements cease to materialize. This means the system rarely explores to the theoretical maximum depth, as it intelligently conserves computational resources once diminishing returns are detected. This design philosophy ensures both efficiency and solution quality, making the choice of 3 transitions both theoretically sound and practically optimal.
>
> ### LLM-as-a-Judge Usage Statistics
>
> We agree this is important. As shown in the DAG Quality Assessment Metrics table above, the LLM-as-a-Judge Selection Rates are **consistently high across datasets (92.5% on MATH, 95.8% on GSM8K, 83.1% on MBPP, and 91.5% on LongBench)**, demonstrating effective quality control and helping with cost control. We will add these statistics (interruption frequency distribution and performance impact) to make the method more transparent.

---

> > ### Comment · Area_Chair_u3zS · 2025-08-04
> > **Please respond to the author's rebuttal post**
> >
> > Hi Reviewer VGGy, I see no response letting me know whether or not the rebuttal has changed
> > your opinion. Could you please let me and the authors know by engaging? This process is critical to enabling the (S)ACs to make a decision on this work.
> >
> > --Your AC

---

> > ### Comment · Reviewer_VGGy · 2025-08-05
> > **Useful feedback, concerns remain**
> >
> > Thank you so much for you diligent work and infomative feedback.
> >
> > ## W1
> >
> > > We would much appreciate it if the reviewer could specify which particular concepts led you to consider them unexplained, as this would help us make the paper clearer and better serve readers
> >
> > I was particularly confused by lines 39–43, especially when no future context was provided. For instance, I had no clear understanding of what "states" refer to or how to compute their "test-time complexity." The term "steady progress" also remained unclear to me, as did the logic behind how "converting the current state into a DAG-based reasoning path" and then "using its structure to reduce dependencies" contributes to achieving such progress. These lines introduce numerous concepts—including states, complexity, steady progress, DAG-based reasoning paths, structure, and dependencies—that appear to lack adequate definition. My confusion deepened when the "structure" was referenced in connection with Figure 4 (line 48).
> >
> > While I eventually managed to grasp all these concepts, this clarity only came after reading the entire paper. I believe this constitutes a significant drawback, as it creates unnecessary confusion for readers early on.
> >
> > > Section 3.2 structure: The section actually follows a deliberate progression
> >
> > The subsection is titled "Emerged Atomic Reasoning," with three subsequent paragraphs labeled "Termination Strategy," "Modular Integration," and "Atomic Structure." The connection between these topics remains unclear. For instance, why does the methodological algorithm (which appears to be an empirical design or heuristic) for termination relate to "emerged atomic reasoning"? Similarly, how does the discussion on integrating AoT with existing approaches tie into the concept of emerging reasoning?
> >
> > > Section 4.4 clarity: Section 4.4 provides systematic empirical validation of our test-time scaling framework
> >
> > I think that is done in Section 4.2 with results in Table 1, while Sectino 4.4 is to integrate AoT with existing frameworks with results in Figure 4. Section 4.4 is quite disordered, making it hard to grasp what methods are integrated and how are the performance.
> >
> > Generally, I appreciate the authors' commitment to improving the presentation. However, a substantial revision is needed to ensure the manuscript meets the clarity standards.
> >
> > ## W2
> >
> > Thank you for your clarifications. I do find you detailed explanations help, while the over-claiming issues still exsits. We would suggest providing additional explanations to more clearly connect the main claims to the results shown in Figures 3 and 4, which would help substantiate the paper's core contributions.
> >
> > ## W3
> >
> > Thank you for you very infomative addtional experiments. I think the experiments about DAG generation quality make a lot sense to to me.
> >
> > A key point of confusion remains regarding the transition hyperparameter. The new experiment shows only negligible gains beyond a value of 3, a finding that seems to conflict with the data in Appendix E, which indicates that most reasoning paths require a depth greater than 3. Is the optimal setting of 3 due to an early exit mechanism, or does further question simplification simply not effective?
> >
> > About the LLM-as-a-Judge usage, sorry for the confusion. I was thinking about the termination strategy. How often does the judge interupt the process and apply termination?

---

> > > ### Author Response · Authors · 2025-08-06
> > >
> > > # W1 (Part 2/3)
> > >
> > > - **Regarding the logic of the two-phase approach**: The Two-phase Transition section starting at line 144 explains why decomposition and transition phases achieve progress. Historical information dependencies are complex, as evidenced by the trend toward increasingly complex reasoning structures (from chains to trees to graphs). Decomposing into a DAG models these complex relationships, then the structural information guides the contraction phase to precisely determine which information can be discarded to construct a new self-contained problem with reduced test-time complexity and lighter historical dependencies. Both main experiments and ablation studies confirm the superiority of this two-phase design.
> > >
> > > - **Regarding undefined concepts**: We define the DAG and its temporal dependencies in Equation 4 (later-generated nodes cannot depend on earlier-generated ones due to LLM's autoregressive sampling). Various structure-based reasoning frameworks like CoT, ToT, and GoT have extensively discussed how these classical graph theory concepts apply to LLM reasoning.
> > >
> > > - **Regarding the Figure 4 structure reference**: The second paragraph of our introduction states "current framework-based test-time scaling methods typically rely heavily on retaining extensive historical information." Subsequent reasoning frameworks (CoT, ToT, GoT) each have their own structural assumptions about reasoning trajectories and design corresponding methods to explore within these structure-supported spaces. Our main experiments demonstrate the excellent effectiveness and efficiency of single explorations on Markov reasoning chains. Figure 4's experiments on integration with other methods further extend test-time exploration, increasing exploration of possible Markov reasoning chains. The performance improvements indicate that richer structural information emerges and is retained as the final answer source.
> > >
> > > We believe these concepts are reasonably distributed throughout the paper, grounded in classical theories like Markov processes and graph theory, with appropriate citations to seminal works in the field.
> > >
> > > ## Connection Between Termination Strategy, Modular Integration and Emerged Atomic Reasoning
> > >
> > > Thank you for asking us to clarify this progression. We'll explain by closely referencing Section 3.2:
> > >
> > > **Core Logical Chain:**
> > >
> > > 1. **Termination Strategy → Ensuring Basic Markov Chain Stability**
> > >    - As stated in lines 168-169: "our Markov chain lacks such a fallback due to its memoryless nature. This amplifies the risk of propagating low-quality transitions"
> > >    - The termination strategy ensures only high-quality state transitions are retained through quality-aware mechanisms (lines 171-175: "we introduce a quality-aware termination strategy that acts as a safeguard")
> > >    - This provides the foundation for subsequent Modular Integration—because answer equivalence is preserved, states can serve as entry points for other methods. Our supplementary DAG data in the rebuttal shows very high answer equivalence preservation rates.
> > >
> > > 2. **Modular Integration → Discovering Deeper Structures Through Extended Exploration**
> > >    - Lines 176-178 explicitly state: "each Markov state is constrained to be an equivalently transformed representation of the original question, the reasoning process forms a semantically aligned and fully self-contained sequence"
> > >    - This modularity enables flexible integration with other methods (lines 179-181: "each state within the chain can be independently routed to specialized solvers, subjected to verification procedures, or further embedded into structured reasoning frameworks")
> > >    - Integration expands exploration of possible Markov chains, allowing deeper reasoning patterns to emerge
> > >
> > > 3. **Atomic Structure → Stable Patterns Emerging from Extended Exploration - Why This is "Emerged"**
> > >    - Lines 189-191 describe this phenomenon: "deeper reasoning states tend to converge into irreducible forms, maintaining a stable and relatively low reasoning token count"
> > >    - Lines 190-192 explicitly define: "We refer to these stable forms as atomic structures: indivisible and self-contained representations that require no further decomposition"
> > >    - The key is in lines 193-194: "Importantly, atomicity is not imposed a priori, but emerges naturally as a property discovered throughout the reasoning process" - we did not predefine transitions to atomic states with minimal test-time reduction. The termination strategy provides exploration quality assurance, modular integration provides exploration breadth and depth, and their combination allows this naturally emerging property in the extended Markov reasoning process to be captured through statistical data (the decreasing token counts in final states shown in Figure 4).
> > >
> > > We believe the original text contains the necessary logical connections. Thank you for your feedback, which will help us enrich this content in revision.

---

> > > ### Author Response · Authors · 2025-08-06
> > >
> > > # W2
> > >
> > > Thank you for acknowledging our commitment. For additional clarifications on connections between main claims and results shown in Figures 3 and 4, please see W1.
> > >
> > > # W3
> > >
> > > ## Transition Hyperparameter and Appendix E Relationship
> > >
> > > We emphasize again that "The purpose of Appendix E is to present statistical results to understand the structural characteristics of decomposed questions." The detailed analysis examines DAG structures generated in the initial decomposition step, focusing on structural properties rather than performance outcomes. The figures and text in the appendix do not address the optimal correspondence between structural differences and maximum transition counts. Actual decomposition depth has no direct relationship with required transition counts. A problem requiring many reasoning steps structurally might provide sufficiently advantageous answers in early transitions by directly solving $Q_i$ or its decomposed $G_i$, eliminating the need to continue iterating along $Q_{i+1}$. Conversely, a seemingly shallow problem might be difficult to compress, requiring additional transitions to accomplish what should ideally happen in one transition (i.e., the expectation that one transition eliminates dependencies on one layer of independent nodes without incoming edges).
> > >
> > > ## LLM-as-a-Judge in Termination Strategy
> > > Thank you for clarifying your question. No worries about the confusion—we're happy to explain how the LLM-as-a-Judge functions in our termination strategy.
> > >
> > > The LLM-as-a-Judge is designed specifically for determining whether to continue or terminate the transition process:
> > >
> > > - **When LLM-as-a-Judge selects $Q_{i+1}$**: The transition continues with $Q_{i+1}$ as the new state
> > > - **When LLM-as-a-Judge does NOT select $Q_{i+1}$**: The process terminates at the current state
> > >
> > > Therefore, the termination probability (how often the judge interrupts the process) is simply **1 minus the probability of selecting $Q_{i+1}$**. These are two sides of the same binary decision—knowing one automatically determines the other.
> > >
> > > We have provided the specific probabilities of LLM-as-a-Judge selecting $Q_{i+1}$ in our previous rebuttal, which directly indicate how frequently the judge applies termination in practice.

---

> ### Author Response · Authors · 2025-08-06
>
> # W1 (Part 1/3)
>
> Thank you for your detailed feedback. We appreciate your careful reading and the opportunity to clarify these important concepts.
>
> ## Clarifications on Key Concepts in the Introduction
>
> We appreciate you pointing out which particular concepts led you to consider them unexplained. Let us clarify:
>
> - **Regarding "states"**: We introduce this concept in lines 37-38: "By exploiting the memoryless property of Markov processes, we design the Markovian reasoning process, where each state encapsulates a self-contained problem, thereby significantly reducing historical dependencies." While this is the first appearance of "state" in our paper, it directly follows the classical theoretical modeling of Markov processes, where state is a fundamental concept. The phrase "where each state encapsulates a self-contained problem..." directly reveals part of our definition - states are "encapsulations" of self-contained problems. This self-containment is a fundamental requirement for states in Markov processes; otherwise, it would violate the memoryless property. We provide a complete derivation of these properties in lines 136-143, including why problems are used for encapsulation and the proof of answer-equivalence.
>
> - **Regarding test-time complexity computation**: Our citation [29] references the seminal work "Scaling LLM Test-Time Compute Optimally can be More Effective than Scaling Model Parameters," which explicitly defines "for inference FLOPs, we use $C = 2ND_{inference}$", where $D$ represents model parameters and $N_{inference}$ the total number of tokens generated at inference time. Since $D$ remains constant when performing test-time compute on the same model, many subsequent works simplify this by omitting the coefficient 2 and $D$, using token count directly as the test-time compute metric, as seen in our references [24, 18]: "s1: Simple test-time scaling" and "T1: Advancing Language Model Reasoning through Reinforcement Learning and Inference Scaling." Since test-time performance is primarily about relative comparisons, some works use metrics that have approximately linear relationships with output tokens - for instance, "Forest-of-Thought: Scaling Test-Time Compute for Enhancing LLM Reasoning" uses the number of trees as the x-axis. We adopt reasoning token consumption cost as our metric.
>
> - **Regarding "steady progress"**: As stated in lines 38-40, "The reasoning process is expressed as a sequence of states with progressively reduced test-time complexity ... To ensure steady progress ...", we aim to ensure that the Markovian state transitions directly contribute to the advancement of the reasoning process. However, directly prompting an LLM to generate states with reduced test-time complexity lacks the necessary structural guidance. The paper acknowledges this challenge (lines 144-148): "However, state transitions aiming at test-time reduction remain challenging for LLMs ... This difficulty arises primarily from the complex historical dependencies within reasoning trajectories. To address this issue, we propose a two-phase transition mechanism ..."
>
> Therefore, we adopt a two-phase approach: first decomposing the problem to extract structural information, then contracting based on this structure to create new self-contained states with reduced test-time complexity. This design aligns with the fundamental principle of test-time scaling—extending the reasoning process within controlled limits to achieve better results. We validate this approach through two ablation variants in Section 4.3.
>
>   - **Variant 1: Without Decomposition** - "where the model directly contracts reasoning trajectories from the initial question without constructing a DAG" (lines 249-250)
>
>   - **Variant 2: Without DAG-guided Contraction** - "where decomposition still occurs, but the contraction step does not rely on any structural guidance. In this setting, only the first naturally independent subproblem is separated out" (lines 41-42)
>
>   Figure 3's experimental results clearly show: "both ablations significantly degrade performance, with the second variant causing a more severe drop. This suggests that partial or superficial structural cues can be more harmful than providing none at all" (lines 253-255).
>
>   Therefore, the two-phase process is necessary and provides relative stability. Regarding the "steady progress" description in the introduction, we acknowledge that we should establish earlier connections with subsequent ablation experiments, briefly clarifying how this designed process provides stable performance improvements compared to directly compressing problem complexity. We appreciate your suggestion.
>
> (Continued in Part 2/3...)

---

> ### Author Response · Authors · 2025-08-06
>
> # W1 (Part 3/3)
>
> ## Addressing the Confusion Between Sections 4.2 and 4.4
>
> Regarding your point that systematic empirical validation of our test-time scaling framework is already completed in Section 4.2, we apologize for what appears to be a typographical error. Our intended meaning was that the entire Section 4 provides systematic empirical validation of our test-time scaling framework. As you correctly noted, Section 4.2 indeed demonstrates the efficiency and effectiveness of our framework as a test-time scaling approach. However, Section 4.3's ablation experiments validate the superiority of our transition mechanism design, while Section 4.4 demonstrates how leveraging our framework's modular integration properties enables combination with other designs to further extend test-time computation, achieving remarkable results when computational resources are invested at scale. This exemplifies the beauty of test-time scaling: multiple design elements can contribute to efficiency while providing a solid foundation for subsequent scaling up.
>
> **Regarding your observation about the organization and clarity of Section 4.4**, we acknowledge that the presentation could be improved. Figure 4's left panel y-axis shows scores on the AIME dataset, and the figure caption specifies which methods are integrated with which parts of the Markov chain structure (including states and chains). In **Section 4.4 Scaling Up Analysis**, we detail two integration approaches and two integration targets:
>
> **Integration Approach 1: State Integration (shown as + Markov state)** - "using individual Markov states as integration points—a lightweight and straightforward approach where intermediate states processed by AoT serve as optimized entry points for other reasoning methods" (line 44). Specifically, "The Markov states $Q_i$ generated by AoT represent simplified, yet answer-equivalent reformulations of the original questions, making them ideal entry points for external methods" (line 53). This is also illustrated in Figure 1, where each problem encapsulated in a Markov state can serve as the original problem for other methods (like tree of thoughts and forest of thoughts shown on the right).
>
> **Integration Approach 2: Chain Integration (shown as + Markov chain)** - "Any intermediate state $Q_i$ can act as an entry point $Q_0$ for other methods, ensuring flexible composition while preserving answer equivalence to the original question" (lines 10-11). "each state within the chain can be independently routed to specialized solvers, subjected to verification procedures, or further embedded into structured reasoning frameworks" (line 58). This works because many reasoning frameworks adopt CoT's chain structure where each node depends on all previously generated nodes. For example, in ToT, each thought generation step in CoT is recorded as child node generation. We replace thought generation with Markov state generation—a decomposition-contraction two-phase transition on the current problem.
>
> **Integration Target 1: Tree Searching (shown as ToT/FoT)** - "Beyond single-state integration, the full Markov sequence $\mathcal{Q}$ generated by AoT can provide a structured scaffold for more complex reasoning frameworks, replacing traditional CoT-based structures" (line 55).
>
> **Integration Target 2: Reflective Refinement (shown as + Reflective Refinement)** - Through "verification-based reflection, where transitions $Q_i \rightarrow Q_{i+1}$ are evaluated by an LLM-as-a-judge" to encourage multiple attempts to find desired transition trajectories rather than exiting after one failure and using $Q_i$ or $G_i$ as the final answer source.

---

> ### Comment · Reviewer_VGGy · 2025-08-08
> **Thank you so much**
>
> Thank you for your very diligent and informative responses! They help me a lot in dispelling some misunderstanding and concerns. I really appreciate your engagement and academic rigor.
>
> I'm adjusting my scores as follows:
> - Clarity 1 -> 2, because some of my own misunderstandings about presentation are dispelled.
> - Rating 2 -> 3, because I believe the the added experiments analyzing the DAG generation quality and termination statistics can further validate the effectiveness of the proposed approaches in a fine-grained fashion.
>
> I'm not rasing the scores higher for now because:
> - W1: I think there still exists a gap between the current presentation and the clarity standards, and I'm not able to review the commited revisions.
> - W2: The existing over-claiming issues might compromise the reliability.
> - W3: **Some major concerns are dispelled.** However, considering AoT as a method releying on LLM rewriting and LLM judges, I am still not fully convinced of all the intermediate implementation details, and how do they influence the final results.
>
> **Finally, I want to thank the authors again for their diligent work and in-depth engagement. I believe all the information will be taken into consideration by the ACs.**

---

### Official Review · Reviewer_GRgo · 2025-07-03

**Clarity:** 3
**Significance:** 3
**Originality:** 3
**Rating:** 4
**Confidence:** 3

**Summary:**

This paper introduces "Atom of Thoughts" (AOT), a new reasoning framework for Large Language Models that uses Markovian processes to reduce computational overhead during test-time scaling. Unlike existing methods like Chain-of-Thought that accumulate extensive historical information, AOT decomposes complex reasoning into self-contained "atomic" units that don't rely on previous context, making inference more efficient. The approach can be integrated with various existing reasoning methods and demonstrates consistent performance improvements as computational budgets increase.

**Questions:**

Does the cost in Figure 4 include the LLM-as-judge and the construction and transition of AoT?

**Ethical Concerns:**

["NO or VERY MINOR ethics concerns only"]

**Limitations:**

Authors have discussed their limitations and social impact in the paper

**Quality:**

2

**Strengths And Weaknesses:**

Strengths:
* Different from previous decompositions, this paper defines a memoryless state in a Markov process, where each state only relies on the one step before. The decomposition requires that each sub-question preserve answer equivalence with the original question. AoT uses a two-phase transition to formulate such a memoryless Markov process, which first creates a DAG and then transfers the DAG to
answer an equivalent independent question. To ensure the equivalence, it uses LLM-as-a-judge to evaluate the state transition.
* The paper provides comprehensive experiments to demonstrate the effectiveness of AoT in different tasks and settings. Experimental results on various tasks (math, code, and QA) show that AoT can improve the model performance in different domains. It also includes the integration of AoT to other reasoning frameworks and the investigation of the scaling law from different aspects.


Weaknesses:
* The motivation is not very clear to me. The construction of memoryless state Qi is based on the DAG. If there is a correct DAG decomposition of the original problem Q0, why not solve the problem directly following the dependencies in this DAG? Since each node in this DAG is an individual thought or sub-question, it should be easier to solve.  However, AoT chooses to convert the DAG to a new answer-equivalence question and solve it, which may lead to unnecessary cost by recreating a similar graph structure again (as in ToT and GoT).
* The performance improvement over the baselines is not significant on MATH, GSM8k, MBPP, and AIME in Table 1.
* The reliability of the answer-equivalence of the state transition depends on LLM-as-a-Judge. In other words, the LLM-based judge is used to check the correctness of the new conditions (independent sub-problems or thoughts). Therefore, the baselines such as FoT and ToT should also use the same LLM-as-a-Judge as the reward function.

Others:
* In Table 1,  AFlow's performance based on DeepSeek-V3 on GSM8k is the highest and should be bolded.
* The font of legends in Figures 3 and 4 is too small.

---

> ### Author Rebuttal · Authors · 2025-07-31
>
> We would like to thank you for your insightful comments and suggestions. We provide detailed responses point by point below. We hope that our clarifications, additional experiments, and responses can address your concerns and provide helpful insights for your evaluation.
>
> > ### Further Clarification on LLM-as-a-Judge
>
> Regarding your mention in the strengths about "using LLM-as-a-judge to evaluate the answer equivalence of state transitions," we would like to clarify that our LLM-as-a-judge design ensures post-transition equivalence through its ingenious design, where ensuring answer equivalence is merely an additional benefit rather than the primary judgment criterion.
>
> In practice, our LLM-as-a-judge selects the best answer from the triplet $(Q\_i, G\_i, Q\_{i+1})$ containing solving information as the evaluation standard. This design **does not require additional reasoning costs** to judge answer equivalence, because $Q\_{i+1}$ that fails to satisfy answer equivalence cannot possibly be selected under this evaluation mechanism. This represents an implicit equivalence guarantee mechanism.
>
> > ### W1: Clarification on Motivation
>
> Thank you for pointing out that the motivation needs clearer articulation. We want to clarify that the nodes in the DAG are actually in a solved state, so "solve the problem directly following the dependencies in this DAG" is indeed the actual situation, because structural relationships cannot be separated from the solving trajectory.
>
> For example, when we solve the problem in Figure 1: "There are two possible triangles ABC satisfying AB \= 10, AC \= b \> 10, and sin B \= 3/5. Find the positive difference between the lengths of side BC.", we cannot generate the second sub-question as "Given cosB=answer of subquestion1, solve for the positive difference between the lengths of side BC.", because this neither saves tokens nor follows common paradigms in natural language reasoning, and it would also lose branching cases, since without solving, the fact that cosB has two possible values here would not be obvious.
>
> Regarding the cost of generating graph structures, we must emphasize that our LLM-as-a-judge design ensures that if additional costs are incurred to continue generating graphs, it indicates that the original graph fails to achieve favorable outcomes in the LLM-as-a-judge evaluation. This implies that the LLM's reasoning process deems such cost expenditure necessary to yield direct performance improvements, making it both essential and beneficial.
>
> Furthermore, due to the test-time complexity reduction requirement during the contraction phase, **AoT ensures at least a non-increasing trend in the complexity of graph structures produced in subsequent transitions**. The increased number of transitions arises from the need for further reasoning, but it does not lead to a linear increase in costs by "recreating a similar graph structure again." Instead, it results in a simpler graph to rectify the suboptimal quality of previous graphs, as the contraction process removes nodes without incoming edges or subsequent dependencies.
>
> > ### W2: Regarding Performance Improvements
>
> Regarding the issue of insignificant performance improvements, we clarify that our main experiments (Table 1) utilize the lightest setting: single Markov chain exploration, which is our core design, without resampling nodes and chains as seen in ToT and FoT's tree search approaches. In Section 4.4's scaling analysis, combining reflection and tree search mechanisms (validated in prior works) with Markov chain exploration results in much more significant gains (Figure 4). We have excluded these results from Table 1 to highlight Markov chain exploration as our fundamental contribution, while acknowledging reflection and tree structures as established prior works for further scaling. To address your concerns, we conducted tests with "ToT + Markov chain + Reflective Refinement" (denoted as AoT\* in the table below) on the mentioned datasets, with the results presented as follows:
>
> **Table: Extended Experiments with Test-Time Scaling Integration**
>
> | Dataset | Method | Accuracy (%) | Cost (USD) | Dataset | Method | Accuracy (%) | Cost (USD) |
> | :---- | :---- | :---- | :---- | :---- | :---- | :---- | :---- |
> | MATH | ToT | 82.0 | 0.351 | AIME | ToT | 78.0 | 0.029 |
> | MATH | FoT (n=8) | 81.7 | 7.283 | AIME | FoT (n=8) | 79.0 | 0.597 |
> | MATH | AoT\* | **84.9** | 11.650 | AIME | AoT\*  | **81.2** | 1.853 |
> | GSM8K | ToT | 91.8 | 0.126 | MBPP | ToT | 73.5 | 0.089 |
> | GSM8K | FoT (n=8) | 94.2 | 2.622 | MBPP | FoT (n=8) | 74.8 | 1.845 |
> | GSM8K | AoT\* | **95.1** | 5.850 | MBPP | AoT\*  | **79.1** | 2.650 |
>
> > ### W3: Regarding LLM-as-a-Judge for Baseline Methods
>
> When implementing same-layer node evaluation for ToT and FoT, **we used LLM-as-a-judge with prompt designs aligned as closely as possible with AoT**. Specifically, all methods follow similar principles in LLM-as-a-judge prompt design: selecting the optimal among multiple solving trajectories. For ToT and FoT, these solutions contain reasoning trajectories for the original problem; for AoT, the selection is among triplets $(Q\_i, G\_i, Q\_{i+1})$ generated from a single state transition. This ensures fairness in comparison.
>
> > ### Q1: Regarding Cost Calculation in Figure 4
>
> Regarding the question about "whether the costs in Figure 4 include LLM-as-a-judge and AoT's construction and transformation," we clearly answer: **Yes, completely included**.
>
> To ensure fairness in experimental comparison, we include the complete reasoning process of all methods in the cost calculation:
>
> - **ToT and FoT**: Include costs of all LLM calls for node generation, node evaluation, etc.
> - **AoT**: Similarly includes all costs for DAG generation, two-phase transition, and LLM-as-a-judge selection among triplets $(Q\_i, G\_i, Q\_{i+1})$
>
> This comprehensive cost calculation ensures our comparison results truly reflect the actual computational overhead of each method, demonstrating the rigor and fairness of our experimental design.
>
> > ### Others: Additional Issues
>
> Thank you for pointing out that the AFlow results based on DeepSeek-V3 on GSM8K data in Table 1 should be bolded. After our review, this occurs in the MATH dataset, and we will make the corresponding correction in the revision. Regarding the issue of overly small font sizes in the legends of Figures 3 and 4, we will also increase the font sizes to ensure clear readability.

---

> > ### Comment · Area_Chair_u3zS · 2025-08-04
> > **Please respond to the author's rebuttal post**
> >
> > Hi Reviewer GRgo, I see no response letting me know whether or not the rebuttal has changed
> > your opinion. Could you please let me and the authors know by engaging? This process is critical to enabling the (S)ACs to make a decision on this work.
> >
> > --Your AC

---

> > ### Comment · Reviewer_GRgo · 2025-08-05
> >
> > Thank the authors for the detailed responses, which have solved most of my concerns, especially on the cost calculation and the baseline comparison. I will suggest adding these clarifications and details to the revision.

---

> ### Author Response · Authors · 2025-08-05
> **Follow-up to Reviewer GRgo Regarding Revisions**
>
> Dear Reviewer GRgo,
>
> We sincerely appreciate the time and effort you have dedicated to reviewing our work. We are particularly grateful for your timely response during the discussion period and are pleased to hear that our detailed responses have solved most of your concerns, especially regarding the cost calculation and baseline comparison.
>
> Following your valuable suggestion, we will definitely add these clarifications and details to our revision to ensure the manuscript is comprehensive and clear. If our revisions and discussions indicate the potential for a score adjustment, we would be very grateful for your consideration.
>
> We remain committed to incorporating all of your suggestions to further enhance the quality of our manuscript. Should you have any additional questions or require further clarification on any aspect of our work, we would be happy to provide more information.
>
> Thank you again for your constructive feedback and thoughtful review.
>
> Best regards,
>
> Authors of NeurIPS 12226

---

### Note · Authors · 2025-08-14

We sincerely thank the AC and all reviewers for their thoughtful feedback and discussions. Below, we summarize the key points addressed and the resulting improvements:

## Key Contributions of Our Work

1. Markovian Reasoning Process: Novel framework leveraging memoryless property to minimize historical dependencies in LLM reasoning
2. Two-Phase Transition Mechanism: Decomposition-contraction approach ensuring answer equivalence while reducing test-time complexity
3. Atomic Reasoning Structure: Leveraging structural priors to enable modular integration, exploring atomic states through test-time scaling for performance improvements

## Major Rebuttal Discussions & Resolutions

### 1. DAG Quality & Answer Equivalence Enforcement (Reviewers cbHf, VGGy, jQ8z)
- Quality Metrics: Answer Equivalence >99%, Test-time Reduction 74-82%, LLM-as-a-Judge Selection Rate 83-96% across all datasets
- Enforcement Mechanism: Answer equivalence is implicitly enforced through selection process - transitions failing to maintain equivalence have very low selection probability (<1%)
- Implementation Clarity: Both decomposition and contraction phases execute through single LLM calls, avoiding costly node-by-node sampling
- Detailed Examples: Provided step-by-step DAG transformation examples and mathematical proof case studies

### 2. Method Robustness & Theoretical Understanding (Reviewers cbHf, VGGy)
- Prompt Robustness: Prompt optimization experiments show limited sensitivity (≤0.7% variation, ≤0.1% with scaling)
- Theoretical Clarification: Reached consensus that atomic reasoning represents convergence under LLM capabilities rather than inherent problem properties
- Hyperparameter Analysis: Showed accuracy plateaus to clarify the distinction between structural depth and actual transition requirements

## Reviewer-Specific Improvements

Reviewer GRgo:
- Addressed the concern about "why not directly solve DAG" by clarifying that nodes already contain solved subproblems
- Demonstrated significant performance improvements when integrating our framework with existing test-time scaling methods

Reviewer VGGy:
- Provided detailed explanations of key concepts by referencing original definitions already present in the manuscript
- Explained the deliberate logical chain in Section 3.2 and how the components build upon each other systematically

We believe these clarifications and improvements address the major concerns and provide a more comprehensive understanding of our work.

---

### Decision · Program_Chairs · 2025-09-17

**Decision:**

Accept (poster)

**Comment:**

The paper proposes a method to extract a DAG of dependencies from an overarching task by breaking it down into sub-questions that can be answered "atomically". While many concerns were raised by reviewers regarding the quality of the underlying graphs and why not to solve them directly, the authors appear to have satisfactorily answered these concerns during the rebuttal period. The only reviewer who recommends rejection VGGy has stated that they would not fight for rejection. The other reviewers also appear to be at a consensus that this work could be accepted as the claims match the empirical results and it is well executed. The approach also blends various prior approaches and bring it to the world of LLMs and as such is worth exploring further. I also lean on the side of accepting given room.